

# Vertical information of CO from TROPOMI total column measurements in context of the CAMS-IFS data assimilation scheme

Tobias Borsdorff[1], Teresa Campos[2], Natalie Kille[3,4,5], Rainer Volkamer[3,4,5], and Jochen Landgraf[1]

[1]Netherlands Institute for Space Research, SRON, Leiden, the Netherlands
[2]Atmospheric Chemistry Observations and Modeling Laboratory and Earth Observing Laboratory National Center for Atmospheric Research Boulder, Colorado USA
[3]Department of Chemistry, University of Colorado Boulder, Boulder, Colorado, USA
[4]Cooperative Institute for Research in Environmental Sciences, University of Colorado Boulder, Boulder, Colorado, USA
[5]Department of Atmospheric and Oceanic Sciences, University of Colorado Boulder, Boulder, Colorado, USA

**Correspondence:** T. Borsdorff (t.borsdorff@sron.nl)

**Abstract.** Since 2017 the Tropospheric Monitoring Instrument (TROPOMI) onboard ESA's Copernicus Sentinel-5 satellite (S5-P) has provided the operational Carbon Monoxide (CO) data product with daily global coverage on a spatial resolution of $5.5 \times 7$ km$^2$ ($7 \times 7$ km$^2$ before August 2019). The European Centre for Medium-Range Weather Forecasts (ECMWF) plans to assimilate the retrieved total columns and the corresponding vertical sensitivities in the Copernicus Atmosphere Monitoring Service Integrated Forecasting System (CAMS-IFS) to improve forecasts of the atmospheric chemical composition. The TROPOMI data will primarily constrain the vertical integrated CO field (VCD) of CAMS-IFS but to a lesser extent also its vertical CO distribution as well. For clear-sky conditions, the vertical sensitivity of the TROPOMI CO data product is useful throughout the atmosphere but for cloudy scenes it varies due to cloud shielding and light scattering. To assess the profile information, we deploy a posteriori profile retrieval that combines individual TROPOMI CO column retrievals with different vertical sensitivities to obtain a vertical CO profile that is then a representative average for the chosen spatial and temporal domain. We demonstrate the approach on three CO pollution cases. For the so called "Rabbit Foot Fire" in Idaho on the 12 August 2018, we estimate a CO profile showing the pollution at an altitude of about 5 km in good agreement with airborne in-situ measurements of the Biomass Burning fluxes of trace gases and aerosols (BB-FLUX) field campaign. The distinct CO enhancement in a plume aloft, decoupled from the ground, is sensed by TROPOMI but is not present in the CAMS-IFS model. For a large-scale event, we analyzed the CO pollution from Siberian wildfires that took place from 14 to 18 August 2018. The TROPOMI data is estimating the height of the pollution plume over Canada at 7 km in agreement with CAMS-IFS. However, CAMS-IFS underestimates the enhanced CO vertical column densities sensed by TROPOMI within the plume by more than 100 ppb. Finally, we study the seasonal biomass burning in the Amazon. During the burning season (1 - 15 August 2019) the CO profile retrieved from the TROPOMI measurements agrees well with the one of CAMS-IFS with a similar vertical shape between ground and 14 km altitude. Hence, our results indicate that assimilating TROPOMI CO retrieval with different vertical sensitivities e.g., under clear-sky and cloudy conditions provide information about the vertical distribution of CO.



# 1 Introduction

The atmospheric trace gas Carbon Monoxide (CO) is predominantly emitted to the atmosphere by incomplete combustion (e.g., due to biomass burning, industrial activity, and traffic). Its only sink is the oxidation reaction with the hydroxyl radical (Spivakovsky et al., 2000), the reaction rate of which determines its moderate atmospheric residence time that varies from days to several weeks (Holloway et al., 2000) depending on latitude and the solar illumination. In combination with its relatively low atmospheric background, CO is widely established to pinpoint pollution hotspots world-wide and to trace the transport of air pollution within the atmosphere e.g., (Gloudemans et al., 2009; Pommier et al., 2013; Schneising et al., 2019; Borsdorff et al., 2020).

One of the primary targets of the Tropospheric Monitoring Instrument (TROPOMI) on ESA's Sentinel-5 satellite (S5-P) is to measure the total column concentration of CO, thereafter referred to as columns, with daily global coverage on a high spatial resolution of $5.5 \times 7$ km$^2$ ($7 \times 7$ km$^2$ before 6 August 2019). The instrument was launched on 13 October 2017 and performs measurements in the visible (270-500 nm), the near-infrared (675-775 nm) and the shortwave infrared (2305-2385 nm) (Veefkind et al., 2012). The shortwave infrared CO retrieval software package (SICOR) was developed for the operational processing of TROPOMI and estimates CO total columns under clear-sky over land and cloudy atmospheric conditions over land and oceans. For clear-sky conditions, the retrieved CO product shows a good sensitivity throughout the atmosphere but can be affected in its vertical sensitivity by the presence of clouds. Therefore, the retrieved columns should be interpreted together with the total column averaging kernels (Borsdorff et al., 2018b, 2019) that are supplied the data product. An extensive validation with ground-based measurements of the TCCON network showed that the TROPOMI data product is compliant with the mission requirements (Borsdorff et al., 2018a; Sha et al., 2021) and is released for public usage (https://scihub.copernicus.eu).

Already early in the mission, the TROPOMI CO data set was compared with the CO fields of ECMWF's Copernicus Atmosphere Monitoring Service (CAMS) Integrated Forecasting System (IFS) and an overall good agreement was found (bias $= 3.2 \pm 5.5\%$, correlation $= 0.97$) (Borsdorff et al., 2018b). CAMS-IFS assimilates various trace gas retrievals from different satellite missions and provides daily global forecast of the chemical composition of the atmosphere up to five days ahead with a spatial resolution of about $40 \times 40$ km$^2$. Currently, the total column CO retrievals from the Measurement of Pollution in the Troposphere (MOPITT) instrument that provides a spatial resolution of 22 km at nadir and from the Infrared Atmospheric Sounding Interferometer (IASI) with a spatial resolution of 12 km at nadir are routinely assimilated (Inness et al., 2015). The MOPITT and IASI CO data are retrieved from thermal infrared (TIR) measurements, details about the approach are given in Deeter et al. (2022) and Clerbaux et al. (2009) respectively. The largest sensitivity of CO column retrievals from the TIR are in the mid-troposphere as shown by Deeter et al. (2013) and George et al. (2015). In contrast, the TROPOMI CO data provides a higher spatial resolution and a vertical sensitivity that covers the full atmosphere including the contribution of the planetary boundary layer, making them particularly suitable for detecting surface sources of CO (Landgraf et al., 2016b). ECMWF plans to assimilate the TROPOMI CO columns and their vertical sensitivities (total column averaging kernels) with CAMS-IFS. The





TROPOMI CO data is already monitored since 2018 within the global CAMS-IFS system and the final assimilation will be activated in the next operational upgrade (CY48R1) scheduled for Q2/2023 (Inness et al., 2022).

In this study, we point out the potential benefits of assimilating TROPOMI CO for CAMS-IFS. The vast amount of TROPOMI data will strongly constrain the total column field of CAMS-IFS including the vertical distribution of CO assumed in the model. This is because the CO column sensitivity to the vertical distribution of CO varies for clear-sky and cloud-sky

conditions and with the observation geometries. Hence the TROPOMI CO observations effectively probe CO in different altitudes. To demonstrate this, we setup a posteriori profile retrieval that combines individual TROPOMI CO column retrievals with different vertical sensitivities and obtains a vertical concentration profile of the CO. We evaluate the approach on three CO pollution events for which we compare the retrieved vertical CO profiles and total columns of TROPOMI with collocated CAMS-IFS simulations that does not assimilate the TROPOMI data yet.

Section 2 introduces the data used in our study, section 3 defines the methodology of the posteriori profile retrieval, section 4 demonstrates the approach on the example of three CO pollution cases and section 5 summarizes our findings and provides conclusions.

## 2 Datasets

The TROPOMI CO dataset used in this study is derived from the shortwave infrared measurements of the instrument deploying

the shortwave infrared CO retrieval (SICOR) algorithm that was developed for operational processing of TROPOMI data (Landgraf et al., 2016a). The retrieval settings in this study are identical to the recent reprocessing of ESA (version 02.04.00). The SICOR algorithm accounts for light scattering effects in the atmosphere by retrieving effective cloud parameters (altitude, optical thickness) together with the total column concentrations of CO, $H_2O$, HDO and $CH_4$. The forward simulation of the retrieval deploys the SEOM-IAS (Scientific Exploitation of Operational Missions - Improved Atmospheric Spectroscopy) cross

section database for all trace gases as described by Borsdorff et al. (2019) and the inversion uses the profile scaling approach that scales a reference profile to fit the spectral measurement (Borsdorff et al., 2014). Here, the reference profile is taken from a spatio-temporally resolved atmospheric transport simulations of the TM5 model (Krol et al., 2005). A detailed outline of all settings for the CO retrieval e.g., spectral windows, priori profiles and other auxiliary data are given by Landgraf et al. (2016b). This study limits the analysis to scenes under clear-sky and low-cloud atmospheric conditions by filtering the data using the

quality assurance value ($q > 0.5$). Furthermore, the artificial striping in flight direction found in the TROPOMI CO data was reduced by the posteriori data correction as discussed by (Borsdorff et al., 2019) based on frequency filtering in the Fourier space.

We compare the TROPOMI CO retrievals with the simulated CO fields of the Copernicus Atmosphere Monitoring Service Integrated Forecasting System (CAMS-IFS). CAMS-IFS is an incremental four-dimensional variational (4D-Var) data

assimilation system that minimizes the difference between the modeled fields and observations to obtain the best forecast by adjusting the initial conditions (Inness et al., 2019, 2015). The aim of CAMS-IFS is to forecast the atmospheric composition in near real-time up to five days ahead provided 6-hourly with a spatial resolution of approximately $40 \times 40$ km$^2$. CO total



column measurements from the Measurement of Pollution in the Troposphere (MOPITT) instrument and the Atmospheric Sounding Interferometer (IASI) are assimilated routinely. ECMWF plans to assimilate the TROPOMI CO columns and their

vertical sensitivities (total column averaging kernels) with CAMS-IFS as well. The spatial resolution of TROPOMI is higher than the one of CAMS-IFS and to overcome this ECWMF is averaging the TROPOMI data into so-called "super-observations" before they are included in the CAMS system. The TROPOMI CO data is already monitored since 2018 within the CAMS-IFS system and the final assimilation will be activated in the next operational upgrade (CY48R1) scheduled for Q2/2023 (Inness et al., 2022). In this study we collocate the CAMS-IFS data with TROPOMI by spatial interpolation to the ground pixel of

the satellite (Borsdorff et al., 2018a). We are interested in comparing the CO vertical concentration profile $\rho_{\mathrm{cams}}$ and the total column given by CAMS-IFS with the TROPOMI observations. Therefore, we calculate the total column concentration $c_{\mathrm{cams}}$ accounting for the vertical sensitivity of the TROPOMI CO product given by the total column averaging kernel $a_{\mathrm{col}}$,

$$c_{\mathrm{cams}} = a_{\mathrm{col}}\rho_{\mathrm{cams}} + \epsilon \tag{1}$$

with the TROPOMI measurement noise contribution $\epsilon$. Hence, the total column $c_{\mathrm{cams}}$ can be compared directly with the

TROPOMI observation.

Finally, we aim to compare the vertical CO profiles derived from the TROPOMI data by deploying the posteriori profile retrieval method with the ones of CAMS-IFS using the profile averaging kernels of the posteriori retrieval. As an additional validation source we use airborne in-situ measurements of the Biomass Burning Flux Measurements of Trace Gases and Aerosol (BB-FLUX) campaign. The NCAR/NSF vacuum UV resonance fluorescence instrument measured in-situ carbon

monoxide mixing ratios during BB-FLUX. It is functionally similar to that of Gerbig et al. (1999). The project analyzed among others an optical thick pollution plume caused by the so called "Rabbit Foot Fire" near to Boise in Idaho on the 12 August 2018 and measured a vertical CO concentration profile for this event. The This date and flight are chosen as they specifically planned an underflight of TROPOMI. These in-situ measurements are in an excellent spatial and temporal overlap with TROPOMI measurements. More details about the campaign are given by Rowe et al. (2022). They validated TROPOMI

CO retrievals under high pollution load with air borne measurements and FLEXPART calculations show that even under this extreme polluted conditions TROPOMI CO is fully compliant with the mission requirements of 10 percent precision and 15 percent accuracy, which makes the data an excellent reference for the validation of the retrieved CO profile from the TROPOMI data.

## 3  Methodology

We have set up a posteriori profile retrieval that estimates a vertical concentration profile $x = (x_1, \ldots, x_n)$ of a trace gas on $n$-layers, representative for a selected region and time range, using TROPOMI total column data different vertical sensitivities. The input of the profile retrieval is a set of $m$ retrieved total columns of a trace gas $c = (c_1, \ldots, c_m)$, their precision $e = (e_1, \ldots, e_m)$, and an estimate of the vertical sensitivity for each retrieved column $c_i$ to the profile component $a_{ij} = \frac{\delta c_i}{\delta x_j}$. So, we





obtain the linear relation between the observed columns $c$ and the profile $x$,

$$c = \mathbf{A}x + e \tag{2}$$

with the Matrix $\mathbf{A} = (a_{ij}) = (a_i)$. The total column averaging kernel $a_i$ varies e.g., due to cloud contamination or different observation geometries as shown in Fig. 1. Hence, using the total columns together with their vertical sensitivity, we can invert Eq. (2) to infer a corresponding vertical concentration profile of a trace gas. To this end we setup a posteriori profile retrieval For this purpose, we minimize the Tikhonov cost function

$$x_{\mathrm{ret}} = \min_x \left\{ ||c - \mathbf{A}x||^2_{\mathbf{S}_e} + ||x - x_{apr}||_{\mathbf{R}} \right\} \tag{3}$$

The $m \times m$ matrix $\mathbf{S_e}$ is a diagonal matrix with $\mathbf{S_{e,ij}} = e_i^2$ on the diagonal and $\mathbf{R}$ is the regularization matrix. We choose $\mathbf{R} = \lambda \mathbf{L_1^T L_1}$ here $\mathbf{L_1}$ is a discrete version of the first derivative:

$$\mathbf{L}_1(i,j) = \begin{cases} 1 & \text{if } i = j \\ -1 & \text{if } i = j - 1 \\ 0 & \text{otherwise} \end{cases} . \tag{4}$$

The regularization parameter $\lambda$ balances the two contributions of the cost function shown in Eq. (3) and thus its value is of crucial importance for the inversion. Figure 2 shows an example of the cost terms as function of $\lambda$. We deploy the L-curve method (Hansen and O'Leary, 1993) to find an appropriate value of the regularization parameter that searches for the highest curvature of this functional dependency that is marked in Fig. 2 with a red dot.

In the linear approximation the solution of the profile retrieval becomes

$$x_{\mathrm{ret}} = \mathbf{G}(c - \mathbf{A}x_{apr}) + x_{apr} \tag{5}$$

with the gain matrix

$$\mathbf{G} = (\mathbf{A}^T \mathbf{S}_e^{-1} \mathbf{A} + \mathbf{R})^{-1} \mathbf{A}^T \mathbf{S}_e^{-1}. \tag{6}$$

In general, the CO total columns calculated from the vertical profiles of the posteriori retrieval are in better agreement with the TROPOMI measurements than the priori assumptions based on TM5. For example, the mean difference between the measured TROPOMI CO columns and TM5 of -17,5% is reduced by the posteriori profile retrieval to only -4.3% for the case over the Amazon during the burning season (1 August 2019 to 15 August 2019). Here, the percentage is relative to the TROPOMI measurements.

The vertical sensitivity of the profile retrieval is described by the profile averaging kernel matrix,

$$x_{\mathrm{ret}} = \hat{\mathbf{A}} \left( x_{\mathrm{true}} - x \right) + x_{apr} \tag{7}$$

with

$$\hat{\mathbf{A}} = \mathbf{G}\mathbf{A} \tag{8}$$





$\hat{\mathbf{A}}$ represents the derivative $\hat{\mathbf{A}} = \frac{\partial x_{\mathrm{ret}}}{\partial x_{\mathrm{true}}}$, where its diagonal elements describe the retrieval sensitivity of a state vector element to its true value as shown in Fig. 3. The degree of freedom for signal

$$\mathrm{DFS} = \mathrm{trace}(\hat{\mathbf{A}}) \,, \tag{9}$$

indicates the total number of independent pieces of information.

The TROPOMI CO retrieval uses the profile scaling retrieval that scales a reference profile $x_{ref}$ to get the spectral measurement in agreement with the simulation. To be most comparable with the TROPOMI CO column retrieval we chose a Tikhonov regularization of the first and a vertical profile expressed relative to a reference profile $x_{ref}$ (Borsdorff et al., 2014) .

## 4   Results

We demonstrated the posteriori profile retrieval approach for three example cases. Figure 4(a) shows the TROPOMI CO

columns retrieved from the measurements over Idaho on the 12 August 2018. The good signal to noise ratio of the TROPOMI measurements allow one to perform precise retrievals from individual ground pixels and by that using the full spatial resolution of the instrument. The measurements show elevated CO values caused by pollution outflow from the "Rabbit Foot Fire" that was burning near to Boise in Idaho. The pollution pattern is not reflected by the simulation of CAMS-IFS as shown by Fig. 4(b). This can be either due to missing emissions of the fire in the model or a time delay of the emissions used in the forecast

run of the model. In both cases the assimilation of TROPOMI data in CAMS-IFS can help to improve the issue. Moreover, the background measured by TROPOMI and modeled by CAMS-IFS is in good agreement when excluding elevated CO values (> 2.8e18 molec/cm2) the bias between TROPOMI and CAMS-IFS is 9.05 ppb (10 %)and the Pearson correlation 0.88. The TM5 CO field is a monthly average with a coarse spatial resolution of $3°$ x $2°$ (longitude×latitude) shown in Figure 4(c) and by that cannot reproduce the pollution plume and not model a realistic background CO concentration (Krol et al., 2005). It is

worth mentioning that TM5 serves as the priori for the TROPOMI CO retrieval and by comparing Fig. 4(c) and (a) it becomes clear how much information about CO is extracted by the retrieval from the TROPOMI measurements. For the posteriori profile retrieval we considered the TROPOMI columns that fall within the black dashed square shown in Fig. 4(a). We select only elevated CO columns (>2.8e18 molec/cm2) to get more information about the pollution signature. Figure 7(a) shows the vertical CO profile retrieved by the posteriori profile retrieval (black) that shows enhanced CO concentrations compared to

CAMS-IFS (blue) and TM5 (yellow). Since both models do not account for the fire emissions the averaged model profiles represent CO background conditions. The BB-FLUX in-situ measurements of the same day (red) show a distinct CO plume at an altitude of about 3-4 km, the retrieved profile from the TROPOMI measurements is vertically smoothed since the retrieval only has a DFS of about 1.79. However, when representing the retrieved profile of TROPOMI relative to its TM5 priori (see Fig. 7(b)) a clear plume signature becomes visible with a maximum at slightly higher altitude than the one of BB-FLUX (about

5 km). We also smoothed the BB-FLUX measurements with the averaging kernel of the posteriori profile retrieval, which results in a similar vertical shape as the one derived from the TROPOMI measurements.

Figure 5 demonstrates the advantage of TROPOMI providing CO retrievals with daily global coverage. It allows one to track the transport of pollution within the atmosphere. Figure 5 (a) shows TROPOMI CO columns measured on 14 August



2018; it shows strong biomass burning sources in Siberia with a CO plume over the arctic ocean. Figure 5 (c) depicts the
corresponding TROPOMI CO measurements on 17 August 2018. Here, the CO pollution plume already reached Canada.
Long-range transports like this predominantly take place in higher altitudes of the atmosphere which makes this case especially
interesting for our study. The atmospheric transport shown by TROPOMI and CAMS are in surprisingly good agreement (see
Fig 5 (b,e)) with a bias of 5.32 ppb (7 %) and 3.31 ppb (5 %) and a Pearson correlation of 0.65 and 0.66 for the 14 and
17 August 2018 respectively. However, the CO enhancement over Canada modeled by CAMS is much lower than the one
measured by TROPOMI and nearer to Siberia, CAMS gives CO enhancement which are not seen by TROPOMI. The reason
might be a time mismatch between the real emissions and the ones assumed in the forecast run of CAMS-IFS. Furthermore,
the CO enhancements modeled by CAMS-IFS at the source of the biomass burning in Siberia show also strong deviations from
TROPOMI. The modeled CO does not reflect all enhancements seen by TROPOMI and seem to be more dispersed than the
satellite measurements. This is also reflected by the profile comparison for this case shown in Fig. 8. The vertical CO profile
retrieved by the posteriori profile retrieval from the TROPOMI measurements (black) shows a clear plume structure at an
altitude of 7-8 km. This time the retrieval has higher DFS of 3 which allows us to obtain more vertical information of the trace
gas. The CAMS-IFS profile (blue) shows a very similar shape as the TROPOMI profile but generally the CO enhancement
is lower in agreement with the column comparison we discussed before. The CAMS-IFS CO profile was smoothed with the
averaging kernel of the TROPOMI posteriori profile retrieval for this comparison. The original CAMS-IFS profile shown in
grey shows even a more pronounced plume signature. TM5 (yellow) shows no enhancement at all is more representative for a
background CO profile of the scene.

As a last example, the posteriori profile retrieval is deployed to estimate a vertical CO profile for a longer time series
of TROPOMI CO column measurements. This also shows that the approach is computationally efficient enough to handle
bigger datasets. Figure 6 shows averaged TROPOMI CO columns for (a) between 16 July 2019 to 1 August 2019 before the
biomass burning season and (d) between 1 August 2019 to 15 August 2019 during the biomass burning season. Hence, burning
activities lead to an enhanced CO background concentration but also individual pollution sources can be distinguished. The
agreement between TROPOMI and CAMS-IFS is again good with a bias of 5.07 ppb (8 %) and 5.35 ppb (8 %) and a Pearson
correlation coefficient of 0.97 and 0.97 before and during the biomass burning season. However, CAMS-IFS is missing some
CO enhancement by individual point sources and in general individual pollution sources look more dispersed in the model run.
We deploy the aposteriori profile retrieval for the two time-ranges specified to estimate a vertical CO profile for an unpolluted
and polluted environment. The black dashed square in Figure 6 (d) indicates the region that we used to select all TROPOMI
CO columns for the profile retrieval. The resulting vertical profiles are shown in Figure 9 (black) together with the ones by
TM5 (yellow) and CAMS-IFS (blue) that we smoothed for the comparison with the averaging kernels of the aposteriori profile
retrieval. The DFS of the retrieval before the biomass burning season and during the biomass burning season is 2.5. For the data
before the biomass burning all three profiles in 9(a) agree well represent a typical unpolluted background situation. The profile
of the posteriori retrieval is not deviating much from its priori TM5 profile here. However, for data during the biomass burning
event this is considerable different. The posteriori profile agrees very well with the CAMS-IFS model and even can reproduce





a similar vertical gradient of the CO concentration field modeled by CAMS-IFS. This shows how much information about the vertical distribution of CO is in the TROPOMI data because of the different vertical sensitivities of the measurements.

## 5   Conclusions

In this study, we deploy a posteriori profile retrieval that combines individual TROPOMI CO column retrievals with different vertical sensitivities and obtains a vertical concentration profile of the CO. We test the approach on three CO pollution events for which we compare the retrieved vertical CO profiles and total columns of TROPOMI with collocated CAMS-IFS simulations that does not assimilate TROPOMI CO data yet.

An example for a small-scale pollution event is the so called "Rabbit Foot Fire" in Idaho on the 12 August 2018. Here, a distinct CO pollution plume is sensed by the retrieved CO columns from measurements of only one TROPOMI overpass. CAMS-IFS shows good agreement with the background CO concentration measured by TROPOMI but no pollution plume is present in the model data. The vertical CO profile that we retrieved from the TROPOMI column retrievals within the plume shows the pollution at an altitude of about 5 km, which agrees reasonably well with airborne in-situ measurements of the BB-FLUX campaign registering the maximum at 3-4 km. This even clearly depicts an example for a challenging case to estimate a vertical profile of CO with the posteriori retrieval. Since CAMS-IFS does not include the pollution event the example indicates the potential of TROPOMI data to improve the model on finer spatial-scales.

The transport of CO pollution from wildfires in Siberia to Canada that took place between 14 and 17 August 2018 is an example for a large-scale event. For all days, we find the CAMS-IFS and TROPOMI total columns in good agreement. However the CO enhancement in the pollution plume is generally lower in the CAMS-IFS data than in the TROPOMI observations. The reason for this could be a time mismatch of the assumed emission in the model, because the emissions seem to be higher for the next day. Furthermore, the pollution sources reflected by the CAMS-IFS columns seem to be more dispersed as shown by the TROPOMI data. We retrieved a vertical profile of CO by selecting the TROPOMI column measurements within the pollution plume over Canada on the 17 August 2018. The shape of the profile agrees well with the modeled one by CAMS-IFS but detects the pollution a bit higher up in the atmosphere (7 km compared to 5 km).

An example for a pollution event that spans over a longer time is the seasonal biomass burning in the Amazon region. Here, we analyze the period before (16 July - 1 August 2019) and during the burning (1 August - 15 August 2019). The total columns of TROPOMI and CAMS-IFS agree reasonably well, but point sources in the CAMS-IFS data seem to be to disperse compared to TROPOMI. The vertical CO profile we retrieved before the burning season does not deviate significantly from our priori assumption based on TM5. The profile derived from the TROPOMI measurements during the burning season agrees very well with the CAMS-IFS data showing even a similar vertical gradient between ground and 14 km altitude.

The vast amount of TROPOMI data will strongly constrain the total column field of CAMS-IFS model. Moreover, in this study we conclude that to some extend the vertical distribution of CO assumed in the model can be improved by assimilating the TROPOMI data product. This is due to fact that retrievals under clear-sky and cloud-sky conditions as well as varying observations geometries have distinct vertical sensitivities and by that are effectively probing CO in different altitudes. This



shows the benefit and need to interpret the TROPOMI CO column measurements together with their vertical sensitivities whenever possible. Finally, not only clear-sky measurements are from interested, but also cloud impacted measurement can yield useful remote sensed data.

## 6 Data availability

The TROPOMI CO data set is available for download at https://s5phub.copernicus.eu/dhus/#/home. The in-situ measurements of the BB-FLUX project can be downloaded from https://data.eol.ucar.edu/project/BB-FLUX. The the CAMS-IFS data is available at https://ads.atmosphere.copernicus.eu/#!/home.

*Author contributions.* Tobias Borsdorff, and Jochen Landgraf performed the TROPOMI CO retrieval and data analysis. The BB-FLUX team provided the airborne in-situ measurements. All authors discussed the results and commented on the manuscript.

*Competing interests.* The authors declare no competing interests. R.V. is an associated editor of AMT.

*Disclaimer.* The presented work has been performed in the frame of the Sentinel-5 Precursor Validation Team (S5PVT) or Level 1/Level 2 Product Working Group activities. The results are based on S5P L1B version 1 data.

*Acknowledgements.* The presented material contains modified Copernicus data [2017,2018] The TROPOMI data processing was carried out on the Dutch national e-infrastructure with the support of the SURF Cooperative. R.V. acknowledges financial support from US National
Science Foundation award AGS-1754019 (BB-FLUX project). We thank Kyle J. Zarzana for the fruitful discussion about the BB-FLUX data and his contribution to the project.



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





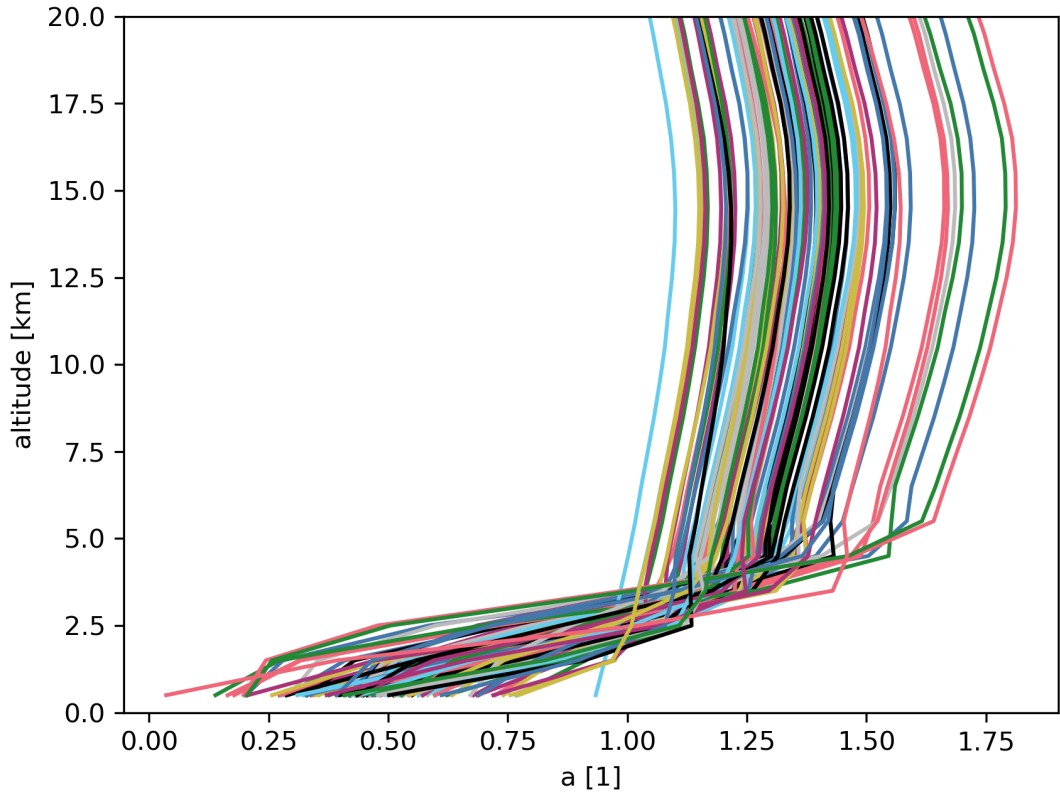

**Figure 1.** Total column averaging kernels of the TROPOMI CO retrieval for different satellite ground pixels selected over the Amazon during the burning season (1 August 2019 to 15 August 2019)

Spivakovsky, C. M., Logan, J. A., Montzka, S. A., Balkanski, Y. J., Foreman-Fowler, M., Jones, D. B. A., Horowitz, L. W., Fusco, A. C., Brenninkmeijer, C. A. M., Prather, M. J., Wofsy, S. C., and McElroy, M. B.: Three-dimensional climatological distribution of tropospheric OH: Update and evaluation, Journal of Geophysical Research: Atmospheres, 105, 8931–8980, https://doi.org/10.1029/1999jd901006, 2000.

Veefkind, J., Aben, I., McMullan, K., Förster, H., de Vries, J., Otter, G., Claas, J., Eskes, H., de Haan, J., Kleipool, Q., van Weele, M., Hasekamp, O., Hoogeveen, R., Landgraf, J., Snel, R., Tol, P., Ingmann, P., Voors, R., Kruizinga, B., Vink, R., Visser, H., and Levelt, P.: TROPOMI on the ESA Sentinel-5 Precursor: A GMES mission for global observations of the atmospheric composition for climate, air quality and ozone layer applications, Remote Sensing of Environment, 120, 70–83, https://doi.org/10.1016/j.rse.2011.09.027, 2012.



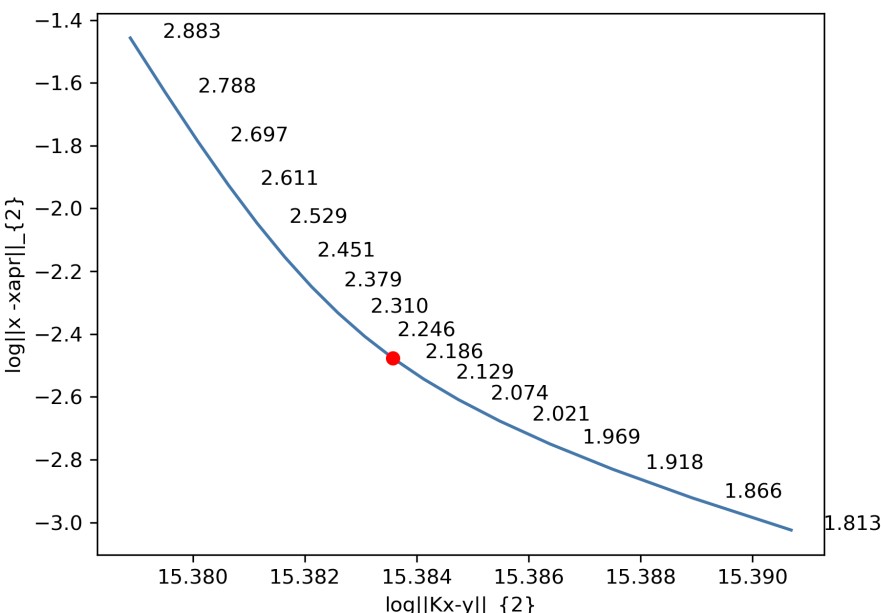

**Figure 2.** L-curve optimization to find the the regularization strength $\alpha$ of the posteriori profile retrieval for the case study over the Amazon during the burning season (1 August 2019 to 15 August 2019). The figure shows the two cost terms as a function of $\lambda$ with the resulting DFS of the posteriori profile retrieval for each data point. The selected optimal point is marked by the red dot.





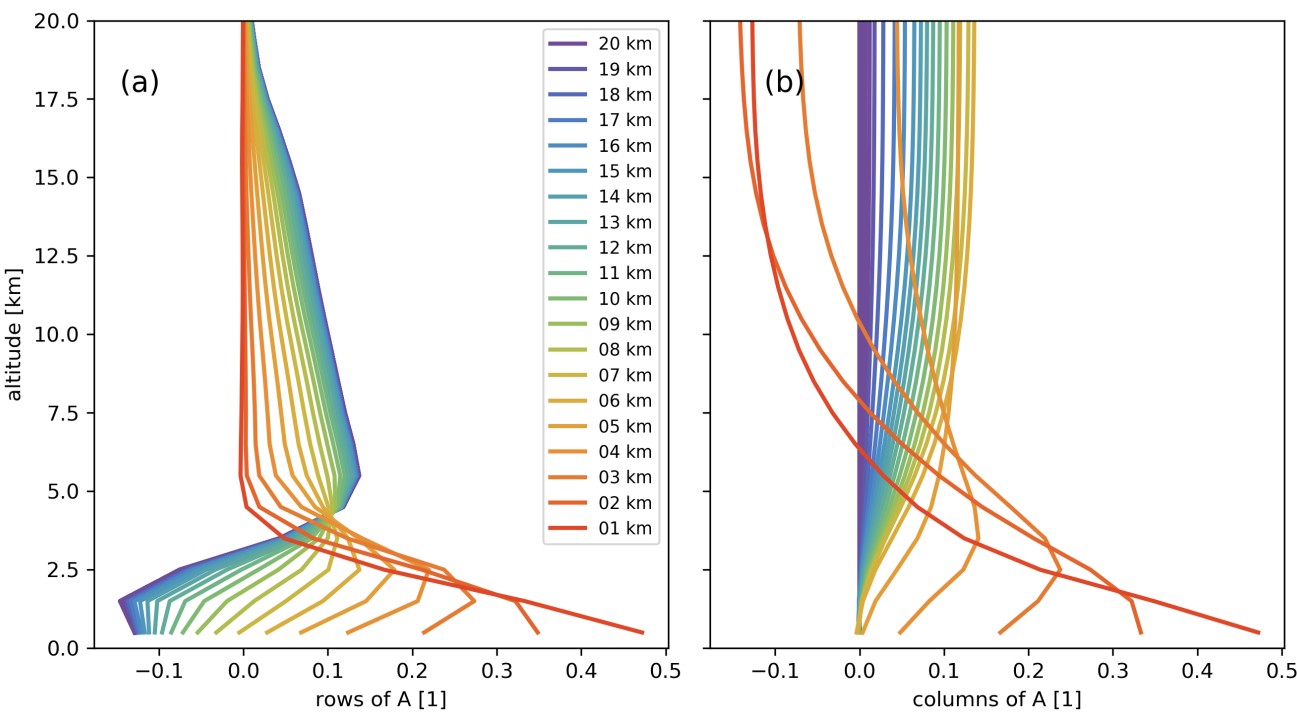

**Figure 3.** Averaging kernel of the posteriori profile retrieval for CO over the Amazon during the burning season (1 August 2019 to 15 August 2019). The left panel (a) shows the rows and the right panel (b) the columns of the averaging kernel matrix. The corresponding altitudes are giving in the legend.

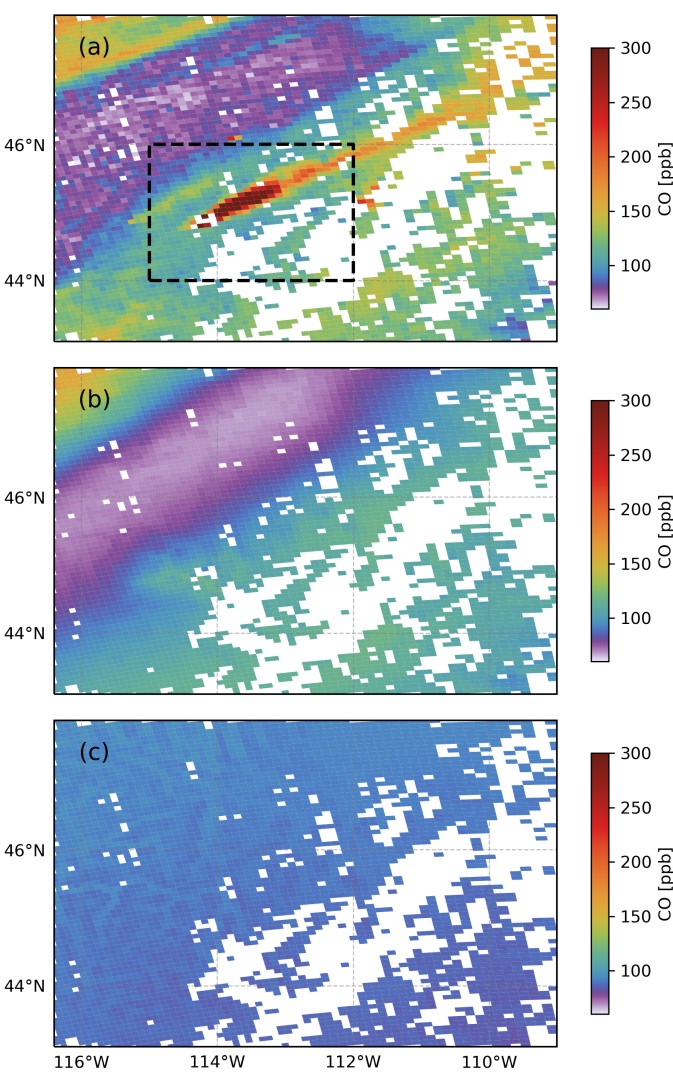

**Figure 4.** CO total columns measured by TROPOMI (a) and the collocated model fields of CAMS-IFS (b) and TM5 (c) are shown. The TROPOMI measurement is taken from orbit 4305 on 12 August 2018 and shows elevated CO caused by the "Rabbit Foot Fire" near to Boise in Idaho not reflected by the data of the two models. The black dashed square marks the region we use for the posteriori profile retrieval.



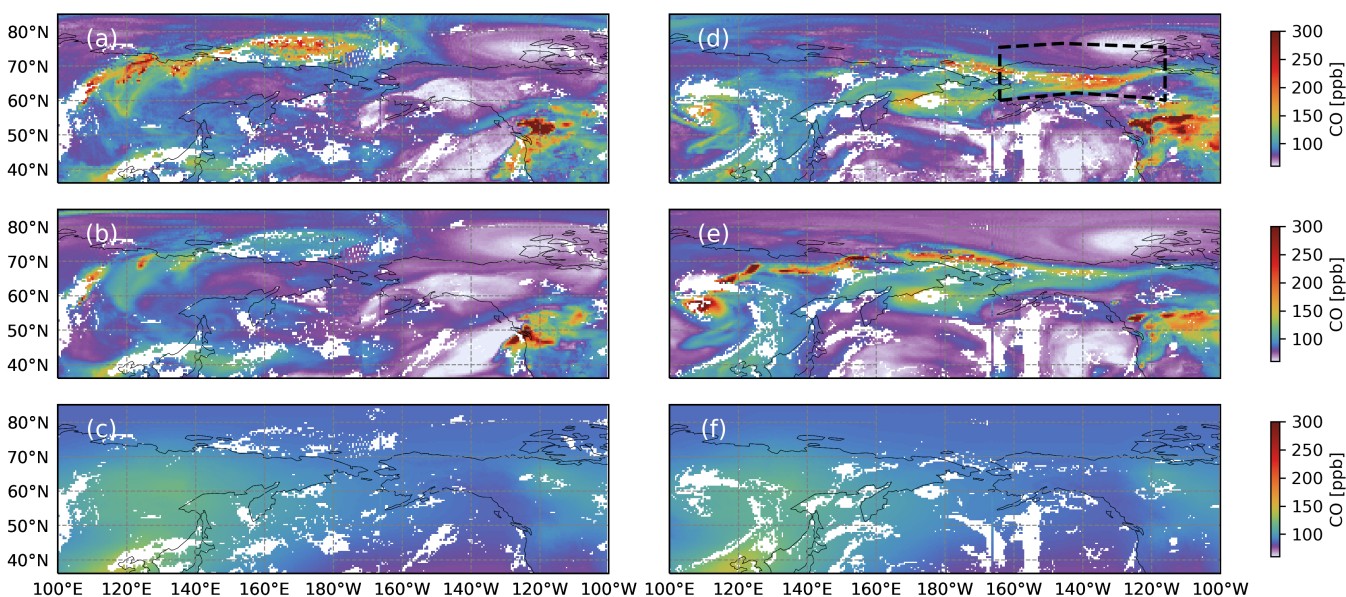

**Figure 5.** Transport of CO pollution from fires in Siberia to Canada is shown on the 14 August 2019 (left column) and 17 August 2019 (right column). TROPOMI total column measurements (a,d) are compared with simulation of CAMS-IFS (b,e) and TM5 (c,f). The black dashed square marks the region we use for the posteriori profile retrieval.

**Figure 6.** Accumulation of CO pollution over the Amazon measured by TROPOMI (a,d) and modeled by CAMS-IFS (b,e) and TM5 (c,f). Left column is the averaged CO field before (16 July to 1 August 2019) and the right column during the burning season (1 August 2019 to 15 August 2019). The black dashed square marks the region we use for the posteriori profile retrieval.





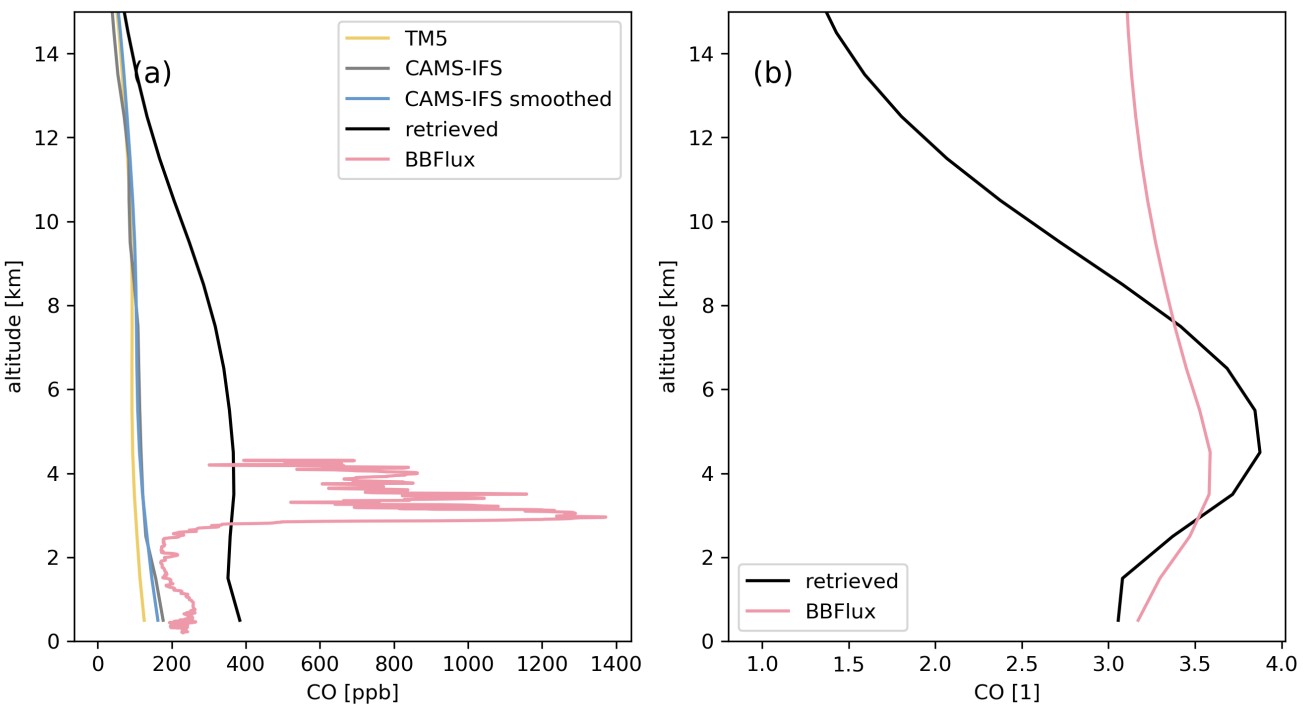

**Figure 7.** Vertical CO profile retrieved from the enhanced TROPOMI CO column measurements on 12 August 2018 caused by the "Rabbit Foot Fire" near to Boise in Idaho (black) in comparison to the one of TM5 (yellow), CAMS-IFS (original data in grey and smoothed with the averaging kernel of the posteriori profile retrieval in blue), and BB-Flux (red). The profiles in panel (a) are given in ppb and in panel (b) in fraction of TM5 which is the priori of the TROPOMI CO retrieval.

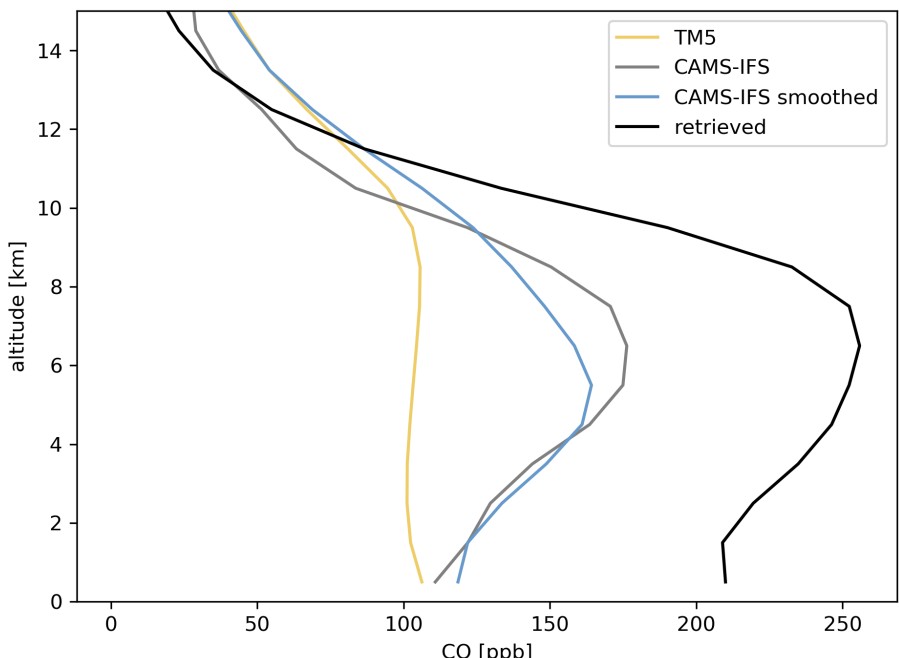

**Figure 8.** Vertical CO profile retrieved from the enhanced TROPOMI CO column measurements on 17 August 2018 caused by pollution transport from Siberia to Canada (black) in comparison to the one of TM5 (yellow), and CAMS-IFS (original data in grey and smoothed with the averaging kernel of the posteriori profile retrieval in blue).



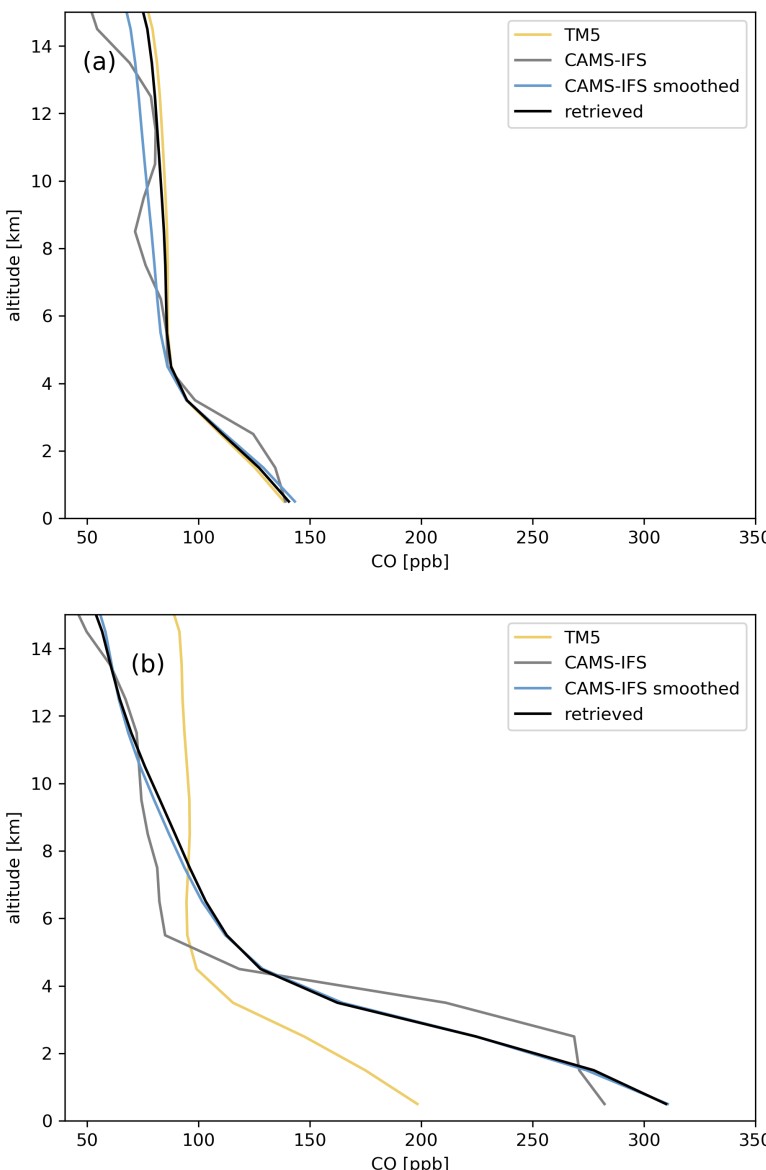

**Figure 9.** Vertical CO profiles retrieved from TROPOMI CO column measurements over the Amazon before (16 July to 1 August 2019) and during the burning season (1 August to 15 August 2019) in comparison with the one of TM5 (yellow), and CAMS-IFS (original data in grey and smoothed with the averaging kernel of the posteriori profile retrieval in blue).