# Peer review of "Vertical information of CO from TROPOMI total column measurements in context of the CAMS-IFS data assimilation scheme"

_Atmospheric Measurement Techniques, 2022_

## Referee Comment (RC2)

**Review of Borsdorff et al. "Vertical information of CO from TROPOMI total column measurements in context of the CAMS-IFS data assimilation scheme"**

This study developed a posteriori profile retrieval method to estimate the mean CO vertical distribution from an ensemble of standard TROPOMI CO column retrievals by incorporating the profile sensitivity from column averaging kernel. The column averaging kernels and retrieval precisions are used to relate the column and the profile, as expressed in equation (2). An optimization scheme was deployed to "retrieve" the mean CO profile. The CO profile is then used to compare with model simulations and field measurements to show that the TROPOMI CO column retrievals contain satisfying vertical distribution information that may not be in the model simulations. For the CAMS-IFS to assimilate TROPOMI CO columns, besides using IASI and MOPITT data, such assessment on the information content from TROPOMI CO column retrieval as reflected in its column averaging kernel is important. Nevertheless, I have two major comments and some specific ones as detailed below. Moreover, the writing of this manuscript, including formatting and simple punctuation, should be improved and double checked through the manuscript.

**Major comments:**

**(1) Assumptions on formulating Equation (2) that relates CO profile and column**
The matrix A is the total column averaging kernel from TROPOMI CO column retrieval. As I understand it:
x_hat – x0 = A*(x - x0) + e
where "x_hat" is the retrieved CO profile (its integration is the retrieved total column "c"); "x0" is the a priori in the retrieval algorithm; "x" is the "truth"; "e" is the error term.
Then,
x_hat + (A-I)*x0 = A*x + e
Since the a priori "x0" or "(A-I)" is not zeros in the TROPOMI column retrieval algorithm, this term cannot be omitted. I would like to know the theoretical basis and assumptions made on formulating equation (2), which is the key equation for the profile retrieval method.

**(2) Effectiveness of the a posteriori profile retrieval method should be evaluated using simulation spectra**
The way to infer the profile from column and column AK is interesting. However, simulation study (so called OSSE) should be carries out to make sure the method works properly. In this way to make sure the retrieval results are not significantly biased, and to quantify the error budget from the a posteriori profile retrieval algorithm.

**Specific comments:**

Line 9: change "individual … retrievals"

Line 11: remove "the" before date.

Line 38: rephrase "are supplied **by** the data product"

Line 50: comma before "respectively"

Line 81: correct the citation format

Line 125: left hand side of Equation (3) should be the cost function value, not x_ret.

Equation (3): Which is used for x_apr in the retrieval algorithm? Is it TM5 simulations? Please add that in the paragraph.

Line 138: in better agreement with TROPOMI CO columns retrieved using profile scaling?

Results section: I would suggest to separate this one big section into 3 parts based on the three different cases.

Line 158: CAMS-IFS assimilates IASI and MOPITT only. The reason the pollution pattern does not show up in the simulation may be because both satellite data failed to capture the anomalies. Please see the images of IASI CO below:

[Figure]

Figure A: (left) IASI CO; (right) MODIS Terra true clor image.
Source 1: https://iasi.aeris-data.fr/co/
Source 2: https://worldview.earthdata.nasa.gov/?v=-124.38027857717674,37.98995560270401,-101.18322922255078,50.78515346100961&t=2018-08-12-T08%3A04%3A01Z

Very likely due to clouds over the fire plume (seen from the MODIS image on the same day), IASI did not make the retrieval. Therefore, it is not reflected in the CAMS-IFS. Adding this background information may help readers to understand the discrepancy between TROPOMI and CAMS-IFS.

Line 159: "This can be either due to missing emissions of the fire in the model or a time delay of the emissions used in the forecast run of the model." You can easily check that by looking into the temporal changes of CO simulations in CAMS-IFS. Or, as I point out above, this is because the IASI data feeded into CAMS-IFS failed to capture the high CO plume.

Line 184, comma before "respectively"

Figure 1:
Do the different colors have meaning? The column averaging kernels differ largely, what are the primary cause? There is an outlier in light blue, which has AK values close to 1.0 for all layers, representing a very ideal case. Why does this one look so peculiar? Also, please rewrite the x-axis label.

Figure 2: please correct the subscript formats for x-axis and y-axis labels

Figure 3: Please explain the difference of the physical meaning of AK columns and rows.

Figure 7-9: Is a way to add error bars for the CO profile retrievals to show if the difference is significant or not?

---

## Author Comment (AC1)

**author comments on the manuscript "Vertical information of CO from TROPOMI total column measurements in context of the CAMS-IFS data assimilation scheme", reviewer 1**

We would like to thank the reviewer for the constructive comments that aided us to improve our manuscript. In this document we provide our replies to the reviewer's comments. The original comments made by the reviewer are numbered and typeset in italic and bold face font. Following every comment, we give our reply. Here line numbers, page numbers and figure numbers refer to the original version of the manuscript, if not stated differently. Additionally, the revised version of the manuscript is added.

**1 Main issues**

1. *In the Rabbit Foot Fire case, one vertical CO profile retrieved from TROPOMI column measurements using the a posteriori profile retrieval method is compared to one collocated airborne vertical profile acquired during the BB-Flux campaign. In each of the other two study cases (Siberia and Amazon fires), one vertical CO profile retrieved from TROPOMI columns is compared to one CAMS-IFS assimilation profile, rather than to actual CO measurements or retrievals. If collocated airborne vertical profiles were unavailable (please discuss in the manuscript if that was the case), such comparison could have been performed using collocated MOPITT and/or IASI vertical profiles. Comparisons to actual measurements/retrievals would be more direct than (and complementary to) comparisons to assimilation results. Please, either compare vertical profiles from TROPOMI and these other two datasets in the manuscript or justify why such comparisons are not performed.*
   **not adjusted**

   We had a serious look at the IASI data that is assimilated by the CAMS-IFS model. A direct comparison between the measurements of TROPOMI and IASI is not straight forward. The attached Fig. 1 to this response shows that the spatial resolution of IASI is much coarser than the one of TROPOMI and that the measurements apparently cannot fully sense the pollution plume seen by TROPOMI. Figure 2 of this response is showing that there is a significant bias between TROPOMI and the IASI retrieval looking at the background CO concentrations but also at the elevated CO locations. Furthermore, we found that there was no data from IASI that could be used to support our application example of the pollution transported from the Siberian fires. Also for the MOPITT data there is not data available for our application example over Canada en Siberia.

   Hence, including this data in our study would mean to address the biases, different timing of measurements, different vertical sensitivities, different spatial resolutions, and the data gaps between the TROPOMI and IASI mission. Furthermore, an explanation would be needed why CAMS-IFS shows a good agreement with TROPOMI and a significant bias with IASI data that is assimilated by the model. This is beyond the scope of our study and would also not support our aim to show the relevance of TROPOMI data for the CAMS-IFS model. We propose that a inter comparison study between TROPOMI, MOPITT, IASI and CAMS-IFS should be done in a separate publication.

2. *In each of the three study sites (Rabbit Foot Fire, Siberia and Amazon fires) the performance of the a posteriori retrieval method is discussed based on a single TROPOMI vertical profile (as well as on one CAMS-IFS and one TM5 profile) per site. To make for a stronger case, please provide multiple examples per site; that would show better the significance of these results.*
   adjusted

   In the manuscript we want to show how much more vertical information can be extracted from the TROPOMI CO column measurements when using more data for each case. Hence, it is not always easy to add more examples. However, we could add an interesting case for the biomass burning in Siberia.

   We added an Subfigure to Figure 7 and changed the caption from :
   "Vertical CO profile retrieved from the elevated TROPOMI CO column measurements on 17 August 2018 ..."
   to "Vertical CO profile retrieved from the elevated TROPOMI CO column measurements on 14 (a) and 17 August 2018 (b) ..."

The caption of Figure 6 is changed from:
"...The black dashed box marks the region we use for the a posteriori profile retrieval. ..." to
"...The black dashed boxes mark the region we use for the a posteriori profile retrieval for the two days. ..." to

We changed the paragraph at P7L82-91 from:
"This is also reflected by the profile comparison for this case shown in Fig. 7. Here one single profile was retrieved from all TROPOMI column measurements that fall within the black dashed box shown in Fig. 6 (d). The black dashed box was chosen empirical to capture the elevated CO values of the event. The vertical CO profile retrieved by the a posteriori profile retrieval from the TROPOMI measurements (black) shows a clear plume structure at an altitude of 7-8 km. This time the retrieval has higher DFS of 3 which allows us to obtain more vertical information of the trace gas. The CAMS-IFS profile (blue) shows similar shape as the TROPOMI profile but generally the CO enhancement is lower in agreement with the column comparison we discussed before. The CAMS-IFS CO profile was smoothed with the averaging kernel of the TROPOMI a posteriori profile retrieval for this comparison. The original CAMS-IFS profile shown in grey shows even a more pronounced plume signature. TM5 (yellow) shows no enhancement at all is more representative for a background CO profile of the scene." to
"This is also reflected by the profile comparison for this case shown in Fig. 7. Here one single profile was retrieved for each of the two days (14 and 17 of August 2018) from all TROPOMI column measurements that fall within the black dashed boxes shown in Fig. 6 (a,d).The black dashed boxes were chosen empirical to capture the elevated CO values of the event. This time the a posteriori retrieval has higher DFS of 2.5 for the 14 of August and 3 for the 17 of August which allows us to obtain more vertical information of the trace gas. The vertical CO profile (black) retrieved by the a posteriori profile retrieval from the TROPOMI measurements on 14 August 2018 shows a clear plume structure at an altitude of 7-8 km with elevated CO levels at the ground (Fig. 7 (a)). The CAMS-IFS profile (blue) shows no plume structure but slightly elevated CO values at the ground. This is changing for the 17 of August 2018 shown in Fig. 7 (b). Here, the vertical profile of the a posteriori retrieval (black) shows a more pronounced plume shape in the same altitude as found on the 14 of August but with lower CO values at the ground. A reason for this could be the atmospheric transport of the pollution away from its source. The CAMS-IFS profile (blue) shows now a similar plume shape as the TROPOMI profile but generally the CO enhancement is lower which is in agreement with the column comparison we discussed before. The CAMS-IFS CO profile was smoothed with the averaging kernel of the TROPOMI a posteriori profile retrieval for this comparison. The original CAMS-IFS profile shown in grey shows even a more pronounced plume signature. TM5 (yellow) shows no enhancement at all and is for both days more representative for a background CO profile of the scene."

3. **CAMS-IFS assimilates vertical CO profiles (and their corresponding AK too? Unclear from the manuscript) from MOPITT and IASI. TROPOMI total CO columns and their AK will also be assimilated by CAMS-IFS starting in Q2/2023.**
   adjusted

   CAMS-IFS is not assimilating vertical profiles of MOPITT and IASI. To make clear that averaging kernels of the data products are used we change the sentence at P2L45 from:
   "...are routinely assimilated."
   to "...are routinely assimilated together with their total column averaging kernels."

**2 Minor issues concerning style / language**

1. **Line 6: "VCD" not used again, please consider removing, for simplicity.**
   adjusted

2. **9, 115, 123, 216: "an a posteriori"**
   adjusted
   In addition we replaced all the occurrences of "posteriori profile retrieval" to "a posteriori profile retrieval".

3. **14: for context, what are the horizontal dimensions of the plume?**
   adjusted

   We change the sentence P1,L13 from:
   "...in a plume aloft, ..." to
   "...in a plume aloft (length=212 km, width=34 km), ..."

4. **17-18: "CAMS-IFS underestimates the enhanced CO vertical column densities sensed by TROPOMI within the plume by more than 100 ppb". What do MOPITT and IASI show regarding total column values? Would aerosols affect differently the SWIR (TROPOMI) and the TIR (MOPITT, IASI, and, thus, CAMS-IFS results)? Please address in the main body of the manuscript.**
   not adjusted

   The focus of this paper is not to compare IASI, MOPITT, and TROPOMI data. Please see our answer to the first comment of this referee. We see a significant bias between TROPOMI and IASI that is not understood. For TROPOMI we stated in the document that the measurements perform robust under high aerosol load cases as validated by the BB-FLUX team (P5L95).

5. **18: "During the burning season (1 - 15 August 2019)". Wording implies that the burning season lasts only those 15 days.**
   adjusted
   We changed the sentence P1L17 from:
   " During the burning season (1 - 15 August 2019) the CO profile retrieved from the TROPOMI measurements agrees ..."to
   " During the burning season the CO profile retrieved from the TROPOMI measurements (1 - 15 August 2019) agrees ..."

6. **23: "carbon monoxide (CO)"**
   adjusted
   also corrected in the abstract P1L2

7. **24: "Its only sink is the oxidation reaction with the hydroxyl radical". OH is the main sink of CO, but not the only one: soil is another important CO sink (e.g., Cordero et al., 2019, Stein et al., 2014).**
   adjusted

   We changed the sentence P2L24 from:
   " Its only sink is the oxidation reaction with the hydroxyl radical (Spivakovsky et al., 2000), the reaction rate of which determines its moderate atmospheric residence time that varies from days to several weeks (Holloway et al., 2000) depending on latitude and the solar illumination. "to
   " Its main sink is the oxidation reaction with the hydroxyl radical (Spivakovsky et al., 2000), the reaction rate of which determines its moderate atmospheric residence time that varies from days to several weeks (Holloway et al., 2000) depending on latitude and the solar illumination. Another important sink of the trace gas is e.g. the oxidation by soil bacteria (Cordero et al., 2019, Stein et al., 2014). "

8. **28: please place "e..g.," inside the parenthesis.**
   adjusted

9. **32: "spatial resolution of 5.5 × 7 km 2 (7 × 7 km 2 before 6 August 2019)" already in Abstract. To avoid repetitions, please consider removing "(7× 7 km 2 before August 2019)" from Abstract.**
   adjusted

10. **33-34: "and the shortwave infrared (SWIR, 2305-2385 nm)"**
    adjusted

11. **34, 69, 70: Consider using "SWIR" instead of "shortwave infrared"**
    adjusted

12. **38: "that are supplied with the data product"**
    adjusted

13. **40: should "and is released for public usage" be "; thus, the TROPOMI CO dataset was released for public usage"?**

adjusted
We changed the sentence at P2L40 from:
" ... and is released for public usage ... "to
" ... and thus, it was released for public usage ... "

14. *42: consider rewording to "The TROPOMI CO data set was compared early in the mission with"*
adjusted

15. *46: for simplicity, consider rewording to "Currently, total column CO retrievals from MOPITT (Measurement of Pollution in the Troposphere instrument) and IASI (Infrared Atmospheric Sounding Interferometer) are routinely assimilated (Inness et al., 2015). The spatial resolution at nadir of these datasets is near 22 x 22 km2 and 12 x 12 km2, respectively."*
adjusted

16. *53: TIR measurements, which are sensitive to the vertical distribution of CO, are most sensitive in the mid troposphere. Are NIR measurements sensitive to the vertical distribution of CO at all?*
adjusted
We changed the sentence at P2L50 from:
" The largest sensitivity of CO column retrievals from the TIR are in the mid-troposphere as shown by ... "to
" TIR measurements, which are sensitive to the vertical distribution of CO, are most sensitive in the mid troposphere as shown by ... "

17. *59: for simplicity, please consider rewording to "varies for clear and cloudy conditions and with observation geometry".*
adjusted

18. *60: Please justify: "Hence the TROPOMI CO observations effectively probe CO in different altitudes."*
adjusted

We remove the sentence at P3L60 from:
" Hence, the TROPOMI CO observations effectively probe CO in different altitudes. "

19. *61: "we implemented an a posteriori profile retrieval method"*
adjusted

20. *63: what were the collocation criteria?*
adjusted
we change the sentence at P4L94 from:
"In this study we collocate the CAMS-IFS data with TROPOMI by spatial interpolation to the ground pixel of the satellite (Borsdorff et al., 2018)." to
"In this study we collocate the CAMS-IFS data with TROPOMI by temporal and spatial interpolation to the ground pixel of the satellite (Borsdorff et al., 2018)."

21. *64: please reword to "collocated CAMS-IFS simulations that do not assimilate"*
adjusted

22. *65, 81, 101, 102, 137, 139, 154, 166, 169, 175, 190, 194, 197, 211, 212, 226, as well as captions of Fig. 2 (x2), 3, 4, 5, 6, 7, 8, 9: "a posteriori" rather than "posteriori"*
adjusted

23. *Similarly, "priori" should be "a priori" in, for example, lines 78, 138, 165, 173, 211, 239, and caption of Fig. 7. Alternatively, please use "prior" instead of "a priori".*

adjusted

We changed all occurrences of "priori" to "a apriori".

24. **66: "demonstrates the approach using three CO pollution cases"**
adjusted

25. **69: the first sentence in the paragraph repeats information already provided in Section 1.**
adjusted
We remove the first sentence and change the sentence at P3L71 from: " The retrieval settings in this study are identical to the recent reprocessing of ESA (version 02.04.00). "to
" The settings of the TROPOMI CO retrieval in this study are identical to the recent reprocessing of ESA (version 02.04.00). "

26. **79: please define "low cloud" in mofre detail.**
adjusted
We change the sentence at P3L79 from:
"This study limits the analysis to scenes under clear-sky and low-cloud atmospheric conditions by filtering the data using the quality assurance value ($q > 0.5$). "to
"This study limits the analysis to scenes under clear-sky conditions ($\tau < 0.5$ and $z < 500m$) and mid-level clouds ($\tau >= 0.5$ and $z < 5000m$) by filtering the data using the quality assurance value ($q > 0.5$). Here $\tau$ is the cloud optical thickness and $z$ the cloud center height."

27. **83: please simplify to "with the simulated CAMS-IFS CO fields." CAMS-IFS had been defined earlier.**
adjusted

28. **86-87: "The aim of CAMS-IFS is to forecast the atmospheric composition in near real-time up to five days ahead provided 6-hourly with a spatial resolution of approximately 40 × 40 km 2." already in Section 1.**
adjusted
We removed the sentence.

29. **87-90: "CO total column measurements from the Measurement of Pollution in the Troposphere (MOPITT) instrument and the Atmospheric Sounding Interferometer (IASI) are assimilated routinely. ECMWF plans to assimilate the TROPOMI CO columns and their vertical sensitivities (total column averaging kernels) with CAMS-IFS as well." already in Section 1.**
adjusted
We removed the paragraph.

30. **92-94: "The TROPOMI CO data is already monitored since 2018 within the CAMS-IFS system and the final assimilation will be activated in the next operational upgrade (CY48R1) scheduled for Q2/2023 (Inness et al., 2022)." already in Section 1.**
adjusted
We removed the paragraph

31. **98: is "$\epsilon$" provided with each TROPOMI retrievals? how is "$\epsilon$" calculated?**
adjusted
We changed the sentence at P4L98 from:
". . . with the TROPOMI measurement noise contribution $\epsilon$."to
". . . with the TROPOMI measurement noise contribution $\epsilon$ that is provided for each retrieval and is propagated from the measurement noise."

32. **104-105: please reword "carbon monoxide" to "CO".**
adjusted

33. **105: please provide precision and accuracy of the airborne measurements. Only one airborne CO profile is shown (Fig. 7a), were any other relevant CO profiles acquired? Having all results and interpretations for this study case based on a single profile is not very persuasive.**

adjusted

We change the sentence at P4L109 from:
" More details about the campaign are given by Rowe et al. (2022). " to
" More details about the campaign including an error estimation of the in-situ measurements are given by Rowe et al. (2022). "

34. ***106: "an optically thick pollution plume"***
adjusted

35. ***107: "This date"***
adjusted

36. ***108: "These in-situ measurements are in an excellent spatial and temporal overlap with TROPOMI measurements" Please provide collocation details: how many hours and km apart are they?***
adjusted
We changed the sentence at P4L108 from:
"These in-situ measurements are in an excellent spatial and temporal overlap with TROPOMI measurements. " to
"The in-situ measurements where done about 45 minutes before the TROPOMI overpass."

37. ***109: for clarity and simplicity, please consider rewording "They validated TROPOMI CO retrievals [. . . ] of the retrieved CO profile from the TROPOMI data.". Please define FLEXPART.***
adjusted

We changed the paragraph at P4L109 from:
" They validated TROPOMI CO retrievals under high pollution load with air borne measurements and FLEXPART calculations show that even under this extreme polluted conditions TROPOMI CO is fully compliant with the mission requirements of 10 percent precision and 15 percent accuracy, which makes the data an excellent reference for the validation of the retrieved CO profile from the TROPOMI data. "
" They validated TROPOMI CO retrievals under high pollution load using air borne measurements and calculations of the FLEXible PARTicle dispersion model (FLEXPART). The study showed that even under extreme polluted conditions the TROPOMI CO dataset is fully compliant with the mission requirements of 10 percent precision and 15 percent accuracy. This makes the BB-FLUX data an excellent reference for the validation of our a posteriori profile retrieval. "

38. ***115-152: This section may be at times difficult to follow for those who are new to this retrieval process. Please define explicitly each new term as it is introduced and provide sufficient background for all readers to gain a basic understanding of the reasoning behind the a posteriori retrieval method.***
adjusted

We change the equation and description P5L120 from:

"So, we obtain the linear relation between the observed columns $c$ and the profile $x$ ,

$$c = \mathbf{A}x + e \tag{1}$$

with the Matrix $\mathbf{A} = (a_{ij}) = (a_i)$, with i =1...m and j=1...n.."
to
"So, we obtain the linear relation between the observed columns $c$ and the profile $x$ ,

$$c = \mathbf{A}x + (\mathbf{I} - \mathbf{A})x_{apr} + e \tag{2}$$

with the Matrix $\mathbf{A} = (a_{ij}) = (a_i)$, with i =1...m and j=1...n and the a priori profile $x_{apr}$ based on TM5. For profile scaling retrievals as used for the TROPOMI CO data the term $(\mathbf{I} - \mathbf{A})x_{apr} = 0$ (Borsdorff et al., 2014). Hence, we can reduce the equation to

$$c = \mathbf{A}x + e \tag{3}$$

"

We updated the manuscript and defined x$_{apr}$ and referenced to TM5. We add the following sentence at page P5L26:
"Here the function min$_x$ is providing the profile that minimizes the Tikhonov cost function."

We add the following sentence at P6L152: "This special type of a profile retrieval is becoming a profile scaling retrieval when the regularization parameter $\lambda \to \infty$."

39. **115: how many layers? At what altitudes/P levels? What criteria were used to select their number and vertical location?**
adjusted

We add the following sentence at P6L152:
"To be compatible with the total column averaging kernels provided by the TROPOMI CO data product the vertical profiles of the a posteriori profile retrieval are defined on the same vertical grid which consists of n=50 layers with an width of 1 km starting from the surface. "

40. **116: would it be "using TROPOMI total column data with different vertical sensitivities."?**
adjusted

41. **116: The text states that the a posteriori profile is "representative for a selected region and time range". What criteria (box size, location) are used to group those TROPOMI retrievals?**
adjusted

We change the sentence at P5L15-16 from:
"We have set up an a posteriori profile retrieval that estimates a vertical concentration profile $x = (x_1, \ldots, x_n)$ of a trace gas on $n$-layers, representative for a selected region and time range, using TROPOMI total column data with different vertical sensitivities. The input of the profile retrieval..." to
"We developed an a posteriori profile retrieval that estimates a vertical concentration profile $x = (x_1, \ldots, x_n)$ of a trace gas on $n$-layers from an ensemble of total column retrievals chosen e.g. for a region or time range of interest. The approach relies on the assumption that the total column retrievals provide different vertical sensitivities e.g caused by cloud contamination or different observation geometries. Hence, the ensemble of column measurements needs to be chosen to ensure that. The input of the a posteriori profile retrieval ..."

42. **117: How is "e" calculated?**
adjusted

We added the following sentence at P5L18:
"Here, $e_i$ is the noise error of the column retrieval $i$ and is the propagated the measurement noise."

43. **121: please clarify (aij ) = (ai ). Does "i" refers to retrieval and "j" to vertical layer?**
adjusted

We add the following sentence at P5L21:
"..., with i =1...m and j=1...n."

44. **From line 97: acol=total column averaging kernel for the TROPOMI total column retrievals. And from line 121: ai=total column averaging kernel. Please clarify what is the difference between both.**
adjusted
There is no difference and accordingly we change the equation P4L97 from:
" ...accounting for the vertical sensitivity of the TROPOMI CO product given by the total column averaging kernel $a_{\mathrm{col}}$,
$$c_{\mathrm{cams}} = a_{\mathrm{col}}\rho_{\mathrm{cams}} + \epsilon \tag{4}$$
" to
" ...accounting for the vertical sensitivity of the TROPOMI CO product given by the total column averaging kernel $a$,
$$c_{\mathrm{cams}} = a\rho_{\mathrm{cams}} + \epsilon \tag{5}$$

"

45. ***129: how is "λ" calculated?***
adjusted

We change the sentence P5L129 from:
"The regularization parameter $\lambda$ balances the two contributions of the cost function shown in Eq. (4) and thus its value is of crucial importance for the inversion. Figure 2 shows an example of the cost terms as function of $\lambda$. We deploy the L-curve method (Hansen and O'Leary, 1993) to find an appropriate value of the regularization parameter that searches for the highest curvature of this functional dependency that is marked in Fig. 2 with a red dot. " to
" The regularization parameter $\lambda$ balances the two contributions of the cost function shown in Eq. (4) and thus its value is of crucial importance for the inversion. In this study, we calculate $\lambda$ by deploying the L-curve method (Hansen and O'Leary, 1993) that searches for the highest curvature of the functional dependency between $\lambda$ and the cost term of the retrieval. Figure 2 shows an example of this functional dependency and the highest curvature is marked with a red dot. "

46. ***137-141: Ideally, total column derived from the a posteriori TROPOMI vertical profiles should be equal to the (averaged?) retrieved TROPOMI total columns; please explain why is that not the case. Also, retrieved TROPOMI total column should differ from total column calculated from TN5 profiles, at least in polluted cases; otherwise, the TROPOMI retrieval would be just a duplicate of its a priori. This paragraph states that the a posteriori profile (at least for the one profile in the Amazon – results from more profiles would make for a much stronger case) is closer to the TM5 profile than the TROPOMI retrievals were; that seems to indicate that actual information in the TROPOMI total column from the "standard" retrieval got lost in the a posteriori profile retrieval. Please clarify and discuss in the Conclusions section. Are similar discussions provided for the Siberia and Amazon cases?***
adjusted
We add the following sentence at P5L138:
"We do not achive here a perfect agreement between the TROPOMI CO columns and the columns calculated from the a posteriori profile because we assume for the a posteriori profile retrieval that the real vertical CO profile is constant in time and space what is not the case in reality. "
We add the following sentence at P5L140:
"For all other example cases in this study the difference between the TROPOMI CO columns and the columns calculated from the a posteriori profile retrieval is in absolute below 0.3%. "

47. ***154: The manuscript would be much stronger if TROPOMI a posteriori vertical profiles were compared with respect to other measurements/retrievals, from MOPITT and IASI if airborne profiles are not available.***
**not adjusted**

Please have a look at our response to the first comment of this reviewer. Unfortunately a direct comparison of TROPOMI with MOPITT and IASI retrievals is not straight forward. The subject of our study is not the inter comparision of different satellite products.

48. ***154: "We applied the a posteriori profile retrieval approach to three different cases."***
adjusted

49. ***156: "and by that" is unclear, please reword.***
adjusted
We changed the sentence at P6L156 from:
" . . . and by that using the full spatial resolution of the instrument. "to
" which allows to use the full spatial resolution of the instrument without the need for averaging individual retrievals to reduce the noise. "

50. ***158: "near Boise, Idaho". For context, please discuss horizontal size of plume and place in the context of TROPOMI, MOPITT, IASI spatial resolution and coverage. Any profiles from MOPITT and IASI sampling the plume?***
adjusted

We change the sentence at P6L158 from:

"...caused by pollution outflow ..." to "...caused by pollution outflow (length=212 km, width=34 km) ..."

51. ***162: How was bias calculated? is TROPOMI higher or lower than CAMS-IFS?***
adjusted

We change the sentence at P6L162 from:

"...the bias between TROPOMI and CAMS-IFS ..."to

"...the bias (TROPOMI - CAMS-IFS) ..."

52. ***163: "The TM5 CO field is a monthly average with a coarse spatial resolution of 3 x 2". However, panel 4.c shows color variations in the pixels represented, which appear to be the same size as the pixels in panels a and b, and much smaller than 3x2 degrees. Please clarify.***
adjusted

We added the following sentence at P6L164:

" Hence, the origin of the small scale variation seen in 4(c) is not the TM5 model but the vertical sensitivity of TROPOMI. For this plot we interpolated both models to the time and ground pixel location of TROPOMI and smoothed them with the total column averaging kernels of the satellite. "

53. ***165: "It is worth mentioning that TM5 serves as the priori for the TROPOMI CO retrieval". Please consider moving to "Datasets"***
adjusted

We changed the sentence at P6L165 from:

" It is worth mentioning that TM5 serves as the a priori for the TROPOMI CO retrieval and by comparing Fig. ...

" to

" Since TM5 serves as the a priori for the TROPOMI CO retrieval by comparing Fig. ..."

54. ***165; 206; and captions of Fig. 4, 5, and 6: please consider rewording "black dashed square" to "black dashed box", since those figures are not square.***
adjusted

55. ***168: Fig. 7 is introduced in the text before Fig. 5 and 6.***
adjusted

56. ***173: "when representing the retrieved profile of TROPOMI relative to its TM5 priori" Does that mean that the posteriori TROPOMI profile was divided by the TM5 profile?***
adjusted

We replace "relative to" by "divided by" at P6L173

57. ***176: please add a smoothed version of the BB-Flux profile to Fig. 7a for a more direct comparison to the a posteriori TROPOMI profile. Unclear how the BB-Flux profile shown in 7b was processed.***
adjusted

We added the smoothed version of the BB-Flux profile to Fig. 7a. The calculation of the profile is addressed in our answer to comment xxx of the reviewer.

58. ***175-176: "We also smoothed [...] TROPOMI measurements" is the smoothed BB-Flux profile shown somewhere?***
adjusted

We added the smoothed BB-Flux profile to Fig.7 (a) following comment xxx of this referee.

59. ***179: "Arctic Ocean"***
adjusted

60. ***179: should it be "Figure 5 (d)" ?***
adjusted

61. **182: consider rewording "surprisingly good agreement" to "remarkably good agreement" or similar.**
    adjusted

62. **183-184: "with a bias of [. . . ] respectively". Are those stats for pixels inside the box only or for the entire region mapped? How was bias calculated, is TROPOMI higher or lower than CAMS?**
    adjusted
    We add the following sentence at P7L184:
    "(the statistics were calculated for the whole map shown, TROPOMI - CAMS-IFS)"

63. **185: what's the time of day of the CAMS-IFS maps? How does it compare to TROPOMI's?**
    **not adjusted**

    CAMS-IFS is provided 6 hourly. This is already stated in the manuscript P3L87. CAMS-IFS data are temporal and spatial interpolated to the TROPOMI ground pixels. This is also already stated in the manuscript P3L94.

64. **186: "a time mismatch between the real emissions and the ones assumed in the forecast run of CAMS-IFS" Please clarify the cause and magnitude of that possible mismatch.**
    adjusted

    We change the sentence at P7L186 from:
    ". . . of CAMS-IFS." to ". . . of CAMS-IFS wich are from the day before."

    We change the sentence at P8L231:
    ". . . assumed emission in the model . . . " to
    ". . . assumed emissions in the model which are taken from the day before, . . . "

65. **192: maybe "shows a similar shape", since they peak at different altitudes and their overall shapes differ quite a bit.**
    adjusted

66. **199: Fig. 6 is introduced after Fig. 8.**
    adjusted

67. **202-203: "with a bias [. . . ] during the biomass burning season". Are those stats for pixels inside the box only or for the entire region mapped? How was bias calculated, i.e., is TROPOMI higher or lower than CAMS?**
    adjusted

    We add the following sentence at P7L203:
    "(the statistics were calculated for the whole map shown, TROPOMI -CAMS-IFS)"

68. **205, 208: "a posteriori"**
    adjusted

69. **210: "agree well and represent"**
    adjusted

70. **214: for consistency, please quantify bias and R in this case too for a posteriori TROPOMI versus CAMS-IFS.**
    **not adjusted**

    The statistics are already given for all cases. Canda: P6L162, Siberia 14.8.2018 and 17.8.2018 P7L183, and Amazon before and after burning: P7L202.

71. **218: collocation criteria? Please provide details early on, in the Datasets section, for example.**
adjusted
This is already done following the xxx comment of this reviewer.

72. **219: please correct to "simulations that do not assimilate"**
adjusted

73. **223: "no pollution plume is present in the model data." Is that because MOPITT and TROPOMI did not capture the plume? Is the plume too small for their spatial resolution? Please explain.**
adjusted

We add the following sentence at P8L223: "We found that for this case no MOPITT data was available and that also the IASI data not captured the plume. Hence, here the CAMS-IFS simulation fully relies on the emissions assumed in the model."

74. **225-226: "This even [. . . ] posteriori retrieval" Please clarify why is this case challenging.**
adjusted

We change the sentence at P8L225 from:
"This even clearly depicts an example for a challenging case to estimate a vertical profile of CO with the a posteriori retrieval. " to
"This example depicts an challenging case to estimate a vertical profile of CO with the a posteriori retrieval because of the limited data availalble for a single event. "

75. **229: "all days" or "both days"?**
adjusted

76. **231-232: please clarify "because the emissions seem to be higher for the next day."**
adjusted
We change the sentence at P8L229 from:
"The reason for this could be a time mismatch of the assumed emission in the model, because the emissions seem to be higher for the next day. " to
"The reason for this could be a time lag of the assumed emission in the model, since the elevated CO at the source region seem to be higher in CAMS-IFS compared to the TROPOM data on 17 August 2019."

77. **243: "to some extent"**
adjusted

78. **244: "due to the fact", "cloudy conditions"**
adjusted

79. **247: consider rewording to "not only clear-sky measurements are valuable" or similar.**
adjusted

80. **Fig. 1: axis label "a[1]" unclear, does it mean "TROPOMI total column averaging kernel [unitless]"?**
adjusted
We add to the caption of Fig. 1:
". . . in unitless representation [1]."

81. **Fig. 2: Are the axis labels correct (please explain) or are the "$_2$" formatting commands? Please explain what are "Kx" and "y". Also, please correct "to find the the regularization" Caption: is "$\alpha$" explained elsewhere?**
adjusted

The axis labels were not consistent with our method section. We corrected this and recalculated the figure. Furthermore, we change the caption from:

"L-curve optimization to find the the regularization strength $\alpha$ of the a posteriori profile retrieval for the case study over the Amazon during the burning season (1 August 2019 to 15 August 2019). The figure shows the two cost terms as a function of $\lambda$ with the resulting DFS of the a posteriori profile retrieval for each data point. The selected optimal point is marked by the red dot. " to
"L-curve optimization to find the regularization strength $\lambda$ of the a posteriori profile retrieval for the case study (1 August 2019 to 15 August 2019) over the Amazon within the burning season. The figure shows the two cost terms as a function of $\lambda$ with the resulting DFS of the a posteriori profile retrieval for each data point. The selected optimal point is marked by the red dot. "

82. ***Fig. 3 caption: 1-15 Aug 2019 is probably not the entire burning season; please reword. Also, reword to "altitudes are given in the legend." Please explain what do the AK rows mean. And the AK columns? Please clarify "[1]" in horizontal axes; should they say "... of A [unitless]"?***
adjusted

We change the caption of Fig.3 from:
"Averaging kernel of the a posteriori profile retrieval for CO over the Amazon during the burning season (1 August 2019 to 15 August 2019). The left panel (a) shows the rows and the right panel (b) the columns of the averaging kernel matrix. The corresponding altitudes are giving in the legend." to
"Averaging kernel of the a posteriori profile retrieval for CO over the Amazon (1 August 2019 to 15 August 2019) within the burning season in unitless representation [1]. The left panel (a) shows the rows and the right panel (b) the columns of the averaging kernel matrix. The rows indicate how one level of the retrieved profile is a smoothed version of all level from the true profile while the columns show how one level of the true profile will affect all level of the retrieved profile. Altitudes are giving in the legend. "

83. ***Fig. 4 caption: for clarity, please consider rewording to, for example "CO total columns for 12 August 2018 from TROPOMI orbit 4305 (a), the CAMS-IFS model (b), and the TM5 model (c). TROPOMI measurements show elevated CO from the Rabbit Foot Fire near Boise, Idaho which was not captured by either of the two models. The black dashed box shows the region analyzed in our a posteriori profile retrieval analysis."***
adjusted

84. ***Fig. 4 caption: add information regarding the spatial resolution in each case. Same for Fig. 6. Please clarify in the main text if a single a posteriori TROPOMI profile was obtained from data in the 2x3 degree ($222x333$ $km^2$) box shown in panel 4a. What criteria were used to select the box?***
adjusted

we add the following sentence at the end of the caption of figure 4 and 5:
"The spatial resolution of TROPOMI is $5.5\times7$ km$^2$, of CAMS-IFS $40\times40$, and of TM5 $3\times2$ degree. CAMS-IFS and TM5 are interpolated in time and space to the TROPOMI ground pixels and the model data is smoothed with the TROPOMI total column averaging kernels."

we add the following sentence at the end of the caption of figure 6:
"The spatial resolution of TROPOMI is $5.5\times7$ km$^2$, of CAMS-IFS $40\times40$, and of TM5 $3\times2$ degree. CAMS-IFS and TM5 are interpolated in time and space to the TROPOMI ground pixels and the model data is smoothed with the TROPOMI total column averaging kernels before the data was averaged on a x by x degree grid."

Furthermore we change the sentence at P6L167 from:
"...shown in Fig. 4(a)" to
"...shown in Fig. 4(a) to retrieve one single vertical profile. The black dashed box was empirical chosen to include the major part of the elevated CO."

85. ***The following applies to analyses summarized in Fig. 4, 5, and 6: How many TROPOMI retrievals went into calculating the one a posteriori vertical profile? Were all the TROPOMI retrievals used, or were they filtered (and how)?***
adjusted

We add the following sentence to the caption of Fig.4:
"Here 71 column retrievals ¿ 2.8e18 were selected."

We add the following sentence to the caption of Fig.5:
"For this, on 14 August 1.5k and on 17 August 1.2k column retrievals ¿ 3e18 were selected"

We add the following sentence to the caption of Fig.6:
"For this, 88k column retrievals were selected before and 83k within the burning season."

86. ***The following applies to Fig. 4, 5, and 6: The text states that the spatial resolution of CAMS-IFS is 40 x 40 km² /pixel and that of TM5 is 3 x 2 degrees/pixel (i.e., 333 x 222 km²/pixel). However, in these maps the pixel size of both CAMS-IFS and TM5 appear to be much smaller that that, and about the same size as TROPOMI's ( 5.5 × 7 km²). Please clarify if the maps have been supersampled or similar; maps with the actual spatial resolution of each dataset would be best.***
adjusted
We updated the figure caption of Fig. 4,5, and 6 as stated in our answer to remark 84 of the referee.

87. ***Fig. 6: Please clarify in the main text if a single a posteriori TROPOMI profile was obtained from data in the 10x5 degree ( 1110x555 km²) box shown in panel 6d. What criteria were used to select the box?***
adjusted
We changed the sentence at P7L207 from:
"...the profile retrieval" to
"the a posteriori profile retrieval that estimates one single vertical profile from it. The black dashed box was chosen empirical to include the region of the strongest biomass burning in that year. "

88. ***Fig. 5: Similarly: please clarify in the main text if a single a posteriori TROPOMI profile was obtained from data in the 50x15 degree ( 5600x1700 km²) box shown in panel 5d. What criteria were used to select the box?***
adjusted

We add the following sentence at P7L189:
"Here one single profile was retrieved from all TROPOMI column measurements that fall within the black dashed box shown in Fig. 6 (d). The black dashed box was chosen empirical to capture the elevated CO values of the event. "

89. ***Fig. 7: The in situ BBFlux profile sampled from 4.5 km to 0 km altitude. However, Fig. 7b shows a transformed BBFlux profile getting all the way up to 15 km altitude. How was that accomplished?***
adjusted

We changed the sentence at P6L175 from:

"We also smoothed the BB-FLUX measurements with the averaging kernel of the a posteriori profile retrieval, which results in a similar vertical shape as the one derived from the TROPOMI measurements. " to

" For better comparison we reduced the vertical resolution of the BB-FLUX profile to the one of TROPOMI. This was done by first extending it with the TM5 profile (yellow) shown Fig. 5(a) for higher altitudes and then smoothing it with the total column averaging kernel of TROPOMI, which results in a similar vertical shape as the one derived from the TROPOMI measurements as can be seen in Fig. 5(b).

90. ***Fig. 8 caption: "enhanced TROPOMI CO column measurements", please explain what does "enhanced" mean here.***
adjusted

We replaced the word "enhanced" by "elevated" in the caption of Fig. 7 and 8.

**3  Figures**

[Figure]

Figure 1: CO total columns for 12 August 2018 from TROPOMI orbit 4305 (a), the CAMS-IFS model (b), and IASI on Metop A (c). TROPOMI measurements show elevated CO from the Rabbit Foot Fire near Boise, Idaho which was not captured by the model. The black dashed box shows the region analyzed in our a posteriori profile retrieval analysis. The spatial resolution of TROPOMI is $5.5 \times 7$ km$^2$, of CAMS-IFS $40 \times 40$, and IASI with a approximately 12 km$^2$ footprint. CAMS-IFS is interpolated in time and space to the TROPOMI ground pixels and the model data is smoothed with the TROPOMI total column averaging kernels. For IASI we show original data.

[Figure]

Figure 2: Accumulation of CO pollution over the Amazon measured by TROPOMI (a,d) and modeled by CAMS-IFS (b,e) and IASI on Metop A (c,f). Left column is the averaged CO field before (16 July to 1 August 2019) and the right column within the burning season (1 August 2019 to 15 August 2019). The black dashed box marks the region we use for the a posteriori profile retrieval. The spatial resolution of TROPOMI is $5.5 \times 7$ km$^2$, of CAMS-IFS $40 \times 40$, and IASI has a footprint of approximately 12 km$^2$. CAMS-IFS and TM5 are interpolated in time and space to the TROPOMI ground pixels and the model data is smoothed with the TROPOMI total column averaging kernels before the data was averaged on a 0.5 by 0.5 degree grid. For IASI we averaged the original data on a 0.5 by 0.5 degree grid.

**4 Additional changes**

1. ***A new co-author is added to the manuscript***
   adjusted
   We added Kyle J. Zarzana as an new co-author and removed the reference to him from the acknowledgements.

**References**

Borsdorff, T., Hasekamp, O. P., Wassmann, A., and Landgraf, J.: Insights into Tikhonov regularization: application to trace gas column retrieval and the efficient calculation of total column averaging kernels, Atmospheric Measurement Techniques, 7, 523–535, https://doi.org/10.5194/amt-7-523-2014, URL https://doi.org/10.5194/amt-7-523-2014, 2014.

Borsdorff, T., aan de Brugh, J., Hu, H., Hasekamp, O., Sussmann, R., Rettinger, M., Hase, F., Gross, J., Schneider, M., Garcia, O., Stremme, W., Grutter, M., Feist, D. G., Arnold, S. G., De Mazière, M., Kumar Sha, M., Pollard, D. F., Kiel, M., Roehl, C., Wennberg, P. O., Toon, G. C., and Landgraf, J.: Mapping carbon monoxide pollution from space down to city scales with daily global coverage, Atmospheric Measurement Techniques Discussions, 2018, 1–19, https://doi.org/10.5194/amt-2018-132, URL https://www.atmos-meas-tech-discuss.net/amt-2018-132/, 2018.

Cordero, P. R. F., Bayly, K., Man Leung, P., Huang, C., Islam, Z. F., Schittenhelm, R. B., King, G. M., and Greening, C.: Atmospheric carbon monoxide oxidation is a widespread mechanism supporting microbial survival, The ISME Journal, 13, 2868–2881, https://doi.org/10.1038/s41396-019-0479-8, URL https://doi.org/10.1038/s41396-019-0479-8, 2019.

Hansen, P. C. and O'Leary, D. P.: The Use of the L-Curve in the Regularization of Discrete Ill-Posed Problems, SIAM Journal on Scientific Computing, 14, 1487–1503, https://doi.org/10.1137/0914086, URL https://doi.org/10.1137%2F0914086, 1993.

Holloway, T., Levy, H., and Kasibhatla, P.: Global distribution of carbon monoxide, Journal of Geophysical Research: Atmospheres, 105, 12 123–12 147, https://doi.org/10.1029/1999jd901173, URL https://doi.org/10.1029/1999JD901173, 2000.

Rowe, J. P., Zarzana, K. J., Kille, N., Borsdorff, T., Goudar, M., Lee, C. F., Koenig, T. K., Romero-Alvarez, J., Campos, T., Knote, C., Theys, N., Landgraf, J., and Volkamer, R.: Carbon Monoxide in Optically Thick Wildfire Smoke: Evaluating TROPOMI Using CU Airborne SOF Column Observations, ACS Earth and Space Chemistry, 6, 1799–1812, https://doi.org/10.1021/acsearthspacechem.2c00048, URL https://doi.org/10.1021/acsearthspacechem.2c00048, 2022.

Spivakovsky, C. M., Logan, J. A., Montzka, S. A., Balkanski, Y. J., Foreman-Fowler, M., Jones, D. B. A., Horowitz, L. W., Fusco, A. C., Brenninkmeijer, C. A. M., Prather, M. J., Wofsy, S. C., and McElroy, M. B.: Three-dimensional climatological distribution of tropospheric OH: Update and evaluation, Journal of Geophysical Research: Atmospheres, 105, 8931–8980, https://doi.org/10.1029/1999jd901006, URL https://doi.org/10.1029/1999JD901006, 2000.

Stein, O., Schultz, M. G., Bouarar, I., Clark, H., Huijnen, V., Gaudel, A., George, M., and Clerbaux, C.: On the wintertime low bias of Northern Hemisphere carbon monoxide found in global model simulations, Atmospheric Chemistry and Physics, 14, 9295–9316, https://doi.org/10.5194/acp-14-9295-2014, URL https://acp.copernicus.org/articles/14/9295/2014/, 2014.

---

## Author Comment (AC2)

**author comments on the manuscript "Vertical information of CO from TROPOMI total column measurements in context of the CAMS-IFS data assimilation scheme", reviewer 2**

We would like to thank the reviewer for the constructive comments that aided us to improve our manuscript. In this document we provide our replies to the reviewer's comments. The original comments made by the reviewer are numbered and typeset in italic and bold face font. Following every comment, we give our reply. Here line numbers, page numbers and figure numbers refer to the original version of the manuscript, if not stated differently. Additionally, the revised version of the manuscript is added.

**1 Major comments**

1. ***Assumptions on formulating Equation (2) that relates CO profile and column The matrix A is the total column averaging kernel from TROPOMI CO column retrieval. As I understand it:***
    $x_{hat} - x0 = A * (x - x0) + e$
    ***where "$x_{hat}$" is the retrieved CO profile (its integration is the retrieved total column "c"); "x0" is the a priori in the retrieval algorithm; "x" is the "truth"; "e" is the error term. Then,***
    $x_{hat} + (A - I) * x0 = A * x + e$
    ***Since the a priori "x0" or "(A-I)" is not zeros in the TROPOMI column retrieval algorithm, this term cannot be omitted. I would like to know the theoretical basis and assumptions made on formulating equation (2), which is the key equation for the profile retrieval method.***
    adjusted

    We change the equation and description P5L120 from:

    "So, we obtain the linear relation between the observed columns $c$ and the profile $x$ ,

    $$c = \mathbf{A}x + e \tag{1}$$

    with the Matrix $\mathbf{A} = (a_{ij}) = (a_i)$, with i =1...m and j=1...n.."
    to

    "So, we obtain the linear relation between the observed columns $c$ and the profile $x$ ,

    $$c = \mathbf{A}x + (\mathbf{I} - \mathbf{A})x_{apr} + e \tag{2}$$

    with the Matrix $\mathbf{A} = (a_{ij}) = (a_i)$, with i=1...m and j=1...n and the a priori profile $x_{apr}$ based on TM5. For profile scaling retrievals as used for the TROPOMI CO data the term $(\mathbf{I} - \mathbf{A})x_{apr} = 0$ (Borsdorff et al., 2014). Hence, we can reduce the equation to

    $$c = \mathbf{A}x + e \tag{3}$$

    "

2. ***(2) Effectiveness of the a posteriori profile retrieval method should be evaluated using simulation spectra The way to infer the profile from column and column AK is interesting. However, simulation study (so called OSSE) should be carries out to make sure the method works properly. In this way to make sure the retrieval results are not significantly biased, and to quantify the error budget from the a posteriori profile retrieval algorithm***
   **not adjusted**

    Our approach is an application of the standard inversion theory for profile inversion that is well established, tested, and applied in scientific community for decades (please the see the publications by e.g. Borsdorff et al. (2014), Phillips (1962), Rodgers and Connor (2003), Twomey (1963)). Our idea of combining the total column averaging kernels is also based on the theory of the standard linear inversion. Of course we tested our numerical implementation but we think that a OSSE study would create overhead in our manuscript and would not add anything to the already established inversion theory. The aim of our manuscript is to apply the theory to show that TROPOMI can improve CAMS-IFS. That is a different objective than validating the inversion theory.

**2 Specific comments**

1. ***Line 9: change "individual ... retrievals"***
   adjusted We replace "individual" by "an ensemble of"

2. ***Line 11: remove "the" before date.***
   adjusted We applied that for all dates in the whole document.

3. ***Line 38: rephrase "are supplied by the data product"***
   adjusted We changed the sentence P2L38 form:
   ". . . are supplied the data product." to
   "are supplied with the data product."

4. ***Line 50: comma before "respectively"***
   adjusted

5. ***Line 81: correct the citation format***
   adjusted

6. ***Line 125: left hand side of Equation (3) should be the cost function value, not $x_{ret}$. Equation (3): Which is used for $x_{apr}$ in the retrieval algorithm? Is it TM5 simulations? Please add that in the paragraph.***
   adjusted Please see our answer to the first comment of the referee. We updated the manuscript accordingly and defined $x_{apr}$ and referenced to TM5. We add the following sentence at page P5L26:
   "Here the function $\min_x$ is providing the profile that minimizes the Tikhonov cost function."

7. ***Line 138: in better agreement with TROPOMI CO columns retrieved using profile scaling?***
   adjusted We add the following sentence at P6L152: "This special type of a profile retrieval is becoming a profile scaling retrieval when the regularization parameter $\lambda \to \infty$."

8. ***Results section: I would suggest to separate this one big section into 3 parts based on the three different cases.***
   adjusted

   We divided the results section 4 in three parts:
   4.1 Rabbit Foot Fire in Idaho
   4.2 Pollution transport from Siberia to Canada
   4.3 Seasonal biomass burning in the Amazon

9. ***Line 158: CAMS-IFS assimilates IASI and MOPITT only. The reason the pollution pattern does not show up in the simulation may be because both satellite data failed to capture the anomalies. Please see the images of IASI CO below Very likely due to clouds over the fire plume (seen from the MODIS image on the same day), IASI did not make the retrieval. Therefore, it is not reflected in the CAMS-IFS. Adding this background information may help readers to understand the discrepancy between TROPOMI and CAMS-IFS.***
   adjusted

   We change the sentence at P6L15 from:
   "This can be either due to missing emissions of the fire in the model or a time delay of the emissions used in the forecast run of the model. In both cases the assimilation of TROPOMI data in CAMS-IFS can help to improve the issue. " to
   " This event was not captured by the MOPITT or IASI satellite measurements most probably caused by clouds over the location of the biomass burning. Hence, the prediction of CAMS-IFS for this case fully dependents on the assumed fire emissions in the model. Consequently, a reason for the missing plume in the CAMS-IFS model could be missing emissions or the one day time delay of fire emissions in the CAMS-IFS forecast run. For all cases the assimilation of TROPOMI data in CAMS-IFS can help to improve the issue. "

10. ***Line 159: "This can be either due to missing emissions of the fire in the model or a time delay of the emissions used in the forecast run of the model." You can easily check that by looking into the temporal changes of CO simulations in CAMS-IFS. Or, as I point out above, this is because the IASI data feeded into CAMS-IFS failed to capture the high CO plume.***
    adjusted

    Please see previous comment.

11. *Line 184, comma before "respectively"*
    adjusted

12. *Figure 1: Do the different colors have meaning? The column averaging kernels differ largely, what are the primary cause? There is an outlier in light blue, which has AK values close to 1.0 for all layers, representing a very ideal case. Why does this one look so peculiar? Also, please rewrite the x- axis label.*
    adjusted The outlier is a typical clear-sky total column averaging kernel hat shows full sensitivity through the atmosphere.

    We change the figure caption from:
    "Total column averaging kernels of the TROPOMI CO retrieval for different satellite ground pixels selected over the Amazon during the burning season (1 August 2019 to 15 August 2019)"

    to "Total column averaging kernels of the TROPOMI CO retrieval for different satellite ground pixels (color coded) selected over the Amazon within the burning season (1 August 2019 to 15 August 2019) in unitless representation [1]. Most of the retrievals are cloud contaminated."

13. *Figure 2: please correct the subscript formats for x-axis and y-axis labels*
    adjusted

14. *Figure 3: Please explain the difference of the physical meaning of AK columns and rows.*
    adjusted We changed the caption from:
    " Averaging kernel of the posteriori profile retrieval for CO over the Amazon during the burning season (1 August 2019 to 15 August 2019). The left panel (a) shows the rows and the right panel (b) the columns of the averaging kernel matrix. The corresponding altitudes are giving in the legend. "to
    " Averaging kernel of the a posteriori profile retrieval for CO over the Amazon (1 August 2019 to 15 August 2019) within the burning season in unitless representation [1]. The left panel (a) shows the rows and the right panel (b) the columns of the averaging kernel matrix. The rows indicate how one level of the retrieved profile is a smoothed version of all level from the true profile while the columns show how one level of the true profile will affect all level of the retrieved profile. Altitudes are giving in the legend. "

15. *Figure 7-9: Is a way to add error bars for the CO profile retrievals to show if the difference is significant or not?*
    **not adjusted**

    We can only propagate the noise error of the total column retrievals on the profile. However, due to the large amount of combined retrieval this error bars are becoming to small to be significant. To get an realistic error estimation we would need the spatial and temporal correlation of the errors on the atmospheric input data which is not available. However, such a error analysis should be better done within an assimilation scheme like CAMS-IFS.

**3  Additional changes**

1. *A new co-author is added to the manuscript*
   adjusted
   We added Kyle J. Zarzana as an new co-author and removed the reference to him from the acknowledgements.

**References**

Borsdorff, T., Hasekamp, O. P., Wassmann, A., and Landgraf, J.: Insights into Tikhonov regularization: application to trace gas column retrieval and the efficient calculation of total column averaging kernels, Atmospheric Measurement Techniques, 7, 523–535, https://doi.org/10.5194/amt-7-523-2014, URL `https://doi.org/10.5194/amt-7-523-2014`, 2014.

Phillips, D. L.: A Technique for the Numerical Solution of Certain Integral Equations of the First Kind, Journal of the ACM, 9, 84–97, https://doi.org/10.1145/321105.321114, URL `https://doi.org/10.1145%2F321105.321114`, 1962.

Rodgers, C. D. and Connor, B. J.: Intercomparison of remote sounding instruments, Journal of Geophysical Research: Atmospheres, 108, 4116, https://doi.org/10.1029/2002jd002299, URL `https://doi.org/10.1029%2F2002jd002299`, 2003.

Twomey, S.: On the Numerical Solution of Fredholm Integral Equations of the First Kind by the Inversion of the Linear System Produced by Quadrature, Journal of the ACM, 10, 97–101, https://doi.org/10.1145/321150.321157, URL `https://doi.org/10.1145%2F321150.321157`, 1963.